# Improving Resistance to Noisy Label Fitting by Reweighting Gradient in SAM

## Abstract

Noisy labels pose a substantial challenge in machine learning, often resulting in overfitting and poor generalization. Sharpness-Aware Minimization (SAM), as demonstrated by Foret et al. (2021), improves generalization over traditional Stochastic Gradient Descent (SGD) in classification tasks with noisy labels by *implicitly slowing noisy learning*. While SAM's ability to generalize in noisy environments has been studied in several simplified settings, its full potential in more realistic training settings remains underexplored. In this work, we analyze SAM's behavior at each iteration, identifying specific components of the gradient vector that contribute significantly to its robustness against noisy labels. Based on these insights, we propose SANER (**S**harpness-**A**ware **N**oise-**E**xplicit **R**eweighting), an effective variant that enhances SAM's ability to manage noisy fitting rate. Our experiments on CIFAR-10, CIFAR-100, and Mini-WebVision demonstrate that SANER consistently outperforms SAM, achieving up to an 8% increase on CIFAR-100 with 50% label noise.

## 1 Introduction

The issue of noisy labels due to human error annotation has been commonly observed in many large-scale datasets such as CIFAR-10N, CIFAR-100N (Wei et al., 2022), Clothing1M (Xiao et al., 2015), and WebVision (Li et al., 2017). Over-parameterized deep neural networks, which have enough capacity to memorize entire large datasets, can easily overfit such noisy label data, leading to poor generalization performance (Zhang et al., 2021). Moreover, the lottery ticket hypothesis (Frankle & Carbin, 2019) indicates that only a subset of the network's parameters is crucial for generalization. This highlights the importance of noise-robust learning, where the goal is to train a robust classifier despite the presence of inaccurate or noisy labels in the training dataset.

Sharpness-Aware Minimization (SAM), introduced by Foret et al. (2021), is an optimizer designed to find better generalization by searching for flat minima. It has shown superior performance over SGD in various tasks, especially in classification tasks involving noisy labels Baek et al. (2024). Understanding the mechanisms behind the success of SAM is crucial for further improvements in handling label noise. Chen et al. (2024) explain SAM's generalization within the benign overfitting framework, showing that SAM outperforms SGD by mitigating noise learning in the early training stages and facilitating more effective learning of features. In linear models, Baek et al. (2024) show that SAM more effectively resists fitting noisy examples than SGD through an explicit up-weighting mechanism that preserves strong gradient contributions from clean examples, thus slowing the learning of noisy instances.

Although SAM effectively mitigates the impact of noise on learning compared to SGD, it still overfits to noisy labels in the later stages of training, as evidenced in Andriushchenko & Flammarion (2022); Baek et al. (2024). We reconfirmed this phenomenon by evaluating the noisy accuracy metric, which measures how well the model overfits to noisy examples in each epoch as shown in Figure 1(a). Baek et al. (2024) analyze SAM's explicit up-weighting mechanism under a sample-wise gradient view. However, in neural network settings, Baek et al. (2024) also admitted that this mechanism fails to fully explain SAM's generalization in tasks with label noise. Motivated by these observations, we aim to investigate why SAM slows down noise learning compared to SGD through a weighting mechanism under a component-wise gradient view in realistic training settings, and how to further enhance SAM's performance in the later stages of training.

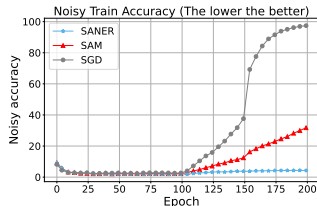 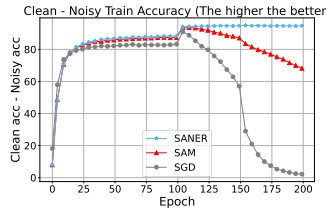 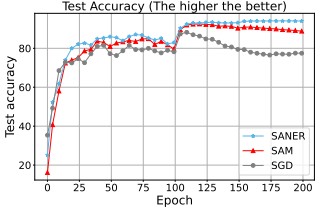

(a) The noisy training accuracy

(b) The gap between clean and noisy training accuracy

(c) The test accuracy

Figure 1: Performance comparison of SAM, SGD, and SANER (ours) trained on ResNet18 with CIFAR-10 under 25% label noise. Noise accuracy indicates how well the model overfits to noisy examples. SAM demonstrates the ability to slow down noisy fitting and increase the gap between clean and noisy accuracy, and our method can further enhance this effect. As a result, SANER outperforms SAM in test accuracy.

In this work, we investigate SAM's capacity to mitigate overfitting to noisy labels and propose an approach to enhance its robustness further, as illustrated in Figure 1. In particular, our investigation reveals two significant findings that motivate our proposed method: (1) During each iteration, specific components in SAM gradient vector contribute significantly to its robustness against label noise. (2) The ratio of these components in noise gradients larger than that in clean gradients, this indicates that further reduction in these components may enhance resistance to noisy label fitting without significantly harming the fitting of clean examples. Building on these insights, we propose a new optimizer, SANER (**S**harpness-**A**ware **N**oise-**E**xplicit **R**eweighting), which explicitly controls noisy fitting more effectively than SAM. This is achieved by further reducing the magnitude of the components in SAM's gradient that correspond to noisy label fitting in each iteration.

Our contributions can be summarized as follows:

- We empirically study the behavior of SAM in component-wise gradients. Specifically, in each iteration, we identify components in the SAM gradient vector that significantly contribute to its resistance against fitting noisy labels. These components have lower magnitudes and the same signs as the corresponding components in SGD. We further analyze their impact on slowing down noisy fitting compared to clean fitting, revealing that reducing the magnitudes of these components has the potential to improve resistance to noisy fitting without significantly harming clean fitting.

- Based on the above idea, we propose SANER, a variant of SAM that has superior resistance to fitting noisy labels compared to SAM. The efficiency of SANER is demonstrated across various datasets, including CIFAR-10, CIFAR-100, and Mini-WebVision, under different settings of noise. SANER consistently outperforms SAM, especially in three challenging overfitting scenarios: increasing model layer width, training without data augmentation, and limited dataset sizes.

- We validate the robustness and efficiency of SANER when integrated with various SAM variants, including ASAM (Kwon et al., 2021), GSAM (Zhuang et al., 2022), FSAM (Li et al., 2024), and VaSSO (Li & Giannakis, 2024). This demonstrates that not only SAM but also its other variants exhibit the characteristics identified in our study.

## 2 BACKGROUND AND RELATED WORKS

### 2.1 SHARPNESS-AWARE MINIMIZATION

Given a dataset $D = (\boldsymbol{x}_i, y_i)_{i=1}^n$ consisting of i.i.d. samples drawn from a population data distribution. Let $f(\boldsymbol{x}_i; \boldsymbol{w})$, parameterized by $\boldsymbol{w} \in \mathbb{R}^d$, represent a neural network, and let $l(f(\boldsymbol{x}_i; w), y_i)$ (shortened as $l_i(\boldsymbol{w})$) denote the loss function between the prediction $f(\boldsymbol{x}_i; \boldsymbol{w})$ and the ground-truth label $y_i$. The empirical training loss is typically defined as:

$$L(\boldsymbol{w}) = \frac{1}{n} \sum_{i=1}^n l_i(\boldsymbol{w}). \tag{1}$$

To minimize this loss, one commonly employs optimization algorithms such as SGD. To enhance generalization performance, SAM (Foret et al., 2021) proposed to seek a flat minimum of the training objective (Equation 1) by minimizing the following robust objective:

$$\min_{\boldsymbol{w}} \max_{||\boldsymbol{\epsilon}||_2 \leq \rho} L(\boldsymbol{w} + \boldsymbol{\epsilon}), \tag{2}$$

where $\rho$ represents the magnitude of the adversarial weight perturbation $\boldsymbol{\epsilon}$. Intuitively, the objective seeks a solution within a neighbor region where the loss remains stable under any $\boldsymbol{\epsilon}$-perturbation. To efficiently optimize this objective, SAM employs a first-order Taylor approximation of the loss, approximating the worst-case $\boldsymbol{\epsilon}$ using the formula as follows:

$$\hat{\boldsymbol{\epsilon}} \approx \arg \max_{||\boldsymbol{\epsilon}||_2 \leq \rho} \boldsymbol{\epsilon}^\top \boldsymbol{g}^{\text{SGD}} = \arg \max_{||\boldsymbol{\epsilon}||_2 \leq \rho} \boldsymbol{\epsilon}^\top \nabla_{\boldsymbol{w}} L(\boldsymbol{w}) = \rho \frac{\nabla_{\boldsymbol{w}} L(\boldsymbol{w})}{||\nabla_{\boldsymbol{w}} L(\boldsymbol{w})||}. \tag{3}$$

Subsequently, the gradient is computed at the perturbed point $\boldsymbol{w} + \hat{\boldsymbol{\epsilon}}$, and the base optimizer (e.g., SGD) with a learning rate $\eta$ is used to update the model parameters in each iteration according to:

$$\boldsymbol{w} = \boldsymbol{w} - \eta \boldsymbol{g}^{\text{SAM}} = \boldsymbol{w} - \eta \nabla_{\boldsymbol{w}} L(\boldsymbol{w}) \Big|_{\boldsymbol{w} + \hat{\boldsymbol{\epsilon}}}. \tag{4}$$

This update guides the model parameters towards a solution robust to perturbations, with only a single extra gradient computation, thereby potentially enhancing generalization.

## 2.2 RELATED WORKS

**SAM**. In addition to the original SAM, several variants have been developed and have shown empirically to improve generalization on datasets with label noise (Kwon et al., 2021; Kim et al., 2022; Jiang et al., 2023; Li & Giannakis, 2024; Li et al., 2024). The convergence of SAM has been studied within the Inexact Gradient Descent framework (Khanh et al., 2023; 2024b), where SAM's perturbed gradient is considered as an approximation of the unperturbed gradient (Khanh et al., 2024a). Several efforts have been made to explain SAM's generalization ability, including investigating SAM's implicit bias (Andriushchenko & Flammarion, 2022), examining the oscillations in SAM's trajectory toward flat minima (Bartlett et al., 2023), exploring how SAM regularizes the eigenvalues of the Hessian of the loss (Wen et al., 2023), and analyzing SAM's generalization through the lens of the bias-variance trade-off (Behdin & Mazumder, 2023).

Shin et al. (2023) examined SAM's performance in overparameterized classification tasks with noisy labels, finding that it yields simpler, flatter solutions than SGD. Baek et al. (2024) attributed SAM's resistance to noisy fitting to gradient up-weighting via the perturbed step mechanism in linear settings. Our work presents a different perspective, showing that SAM also down-weights certain gradient components in realistic settings in each iteration, which helps slow down noisy fitting.

**Label Noise**. Many methods have been developed to enhance noise robustness in deep learning, including (1) Designing loss functions that are less sensitive to noisy examples (Zhang & Sabuncu, 2018; Menon et al., 2020; Ma et al., 2020; Wei et al., 2023); (2) Implementing a sample weighting mechanism ensures that the model prioritizes learning from clean data, reducing the impact of noisy examples during training (Liu & Tao, 2015; Ren et al., 2018; Jiang et al., 2018; Wei et al., 2020); (3) Utilizing regularization techniques to improve generalization in the presence of label noise (Lukasik et al., 2020; Xia et al., 2021; Bai et al., 2021; Liu et al., 2022); and (4) Adopting training strategies based on semi-supervised learning (Nguyen et al., 2020; Li et al., 2020), meta-learning (Ren et al., 2018; Shu et al., 2019; Wei et al., 2020), or self-supervised learning (Li et al., 2022).

Our gradient-based approach focuses on explicitly identifying and reducing components of the gradient vector that contribute more to learning from noisy examples in each iteration. A closely related study is the CDR method by Xia et al. (2021), which isolates noisy (non-critical) and clean (critical) parameters in each iteration based on the magnitude of the product of their gradients and corresponding weights to prevent memorization of the noisy labels. The number of critical parameters is determined by estimating the noise rate in the training data. Unlike their approach, our method does not rely on estimating the noise rate but utilizes SAM's behavior in each iteration to further enhance the slowing down of noisy fitting.

## 3 ANALYZING GRADIENT BEHAVIOR OF SAM

In this section, we empirically demonstrate that, in each iteration, the down-weighted gradient magnitude in SAM contributes significantly to its resistance to label noise. We present various experiments and provide the underlying intuition and motivation for our experiments. We conducted experiments using ResNet18 on CIFAR-10 with 25% label noise, following hyperparameters detailed in Appendix A.1. For a component-wise gradient analysis, we denote $g_i$ as the gradient component corresponding to the $i$-th parameter of gradient vector $\boldsymbol{g}$, and $d$ is the number of parameters in the neural network.

We raise two key questions: (1) How does SAM's component-wise gradient differ from SGD in each iteration? (2) Are there specific component-wise gradients that focus on preventing noisy fitting in each iteration?

**Gradient weighting in SAM.** In each iteration, we categorize each gradient component into three groups based on the ratio of its value in SAM and SGD, defined as $r_i = g_i^{\text{SAM}}/g_i^{\text{SGD}}$, and analyze their proportions during SAM's training process as follows:

**Group A:** SAM increases SGD gradient component. $\mathcal{S}_A = \{i \in \{1, 2, \ldots, d\} \mid r_i \geq 1\}$.

**Group B:** SAM decreases SGD gradient component. $\mathcal{S}_B = \{i \in \{1, 2, \ldots, d\} \mid 0 \leq r_i < 1\}$.

**Group C:** SAM reverses SGD gradient component direction. $\mathcal{S}_C = \{i \in \{1, 2, \ldots, d\} \mid r_i < 0\}$.

Figure 2 illustrates the percentage of three groups of gradient components across all parameters of the neural network model during SAM training. It indicates that Group A, where gradient components are up-weighted, accounts for 50% of the parameters. *Group B, where gradient components are down-weighted, covers around 30-40% of the parameters* during most of the training phase. Group C, where parameters have gradients that are reversed in direction, starts with a small portion but increases towards the end of the training phase. This may be because the model mostly converges and ends up in a rough landscape, causing the backward step to diverge from the direction of SGD.

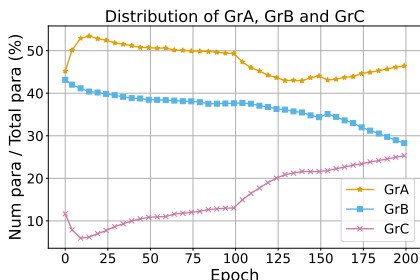

Figure 2: Parameter distribution (%) of groups A, B, and C during training.

Analyzing Group C is particularly challenging due to the inconsistency between the objectives of SAM and SGD. The divergence in gradient component directions complicates the learning process, as reversed gradients may hinder effective learning from the data. Moreover, our study focuses on the "memorization" phase (Arpit et al., 2017), where the transition from fitting clean examples to overfitting noisy examples occurs. This typically happens during the middle stage of training, when most clean examples have already been learned. During this period, Groups A and B are still dominant compared to Group C. Therefore, in this section, we focus on comparing the effects of Group A and Group B of SAM on noisy fitting, leaving the analysis of Group C for future work.

The experiment reveals that SAM not only up-weights gradients but also involves a significant portion of down-weighted gradients. In linear models, Baek et al. (2024) suggested that the up-weighting of gradients (Group A) helps maintain focus on clean examples for a longer period. However, they observed that this mechanism could not explain SAM's ability to slow down noisy fitting in neural networks. We hypothesize that Group B, due to its substantial presence, may play a crucial role in addressing the issue of noisy fitting in realistic training settings. This study investigates the role of Group B in mitigating noisy fitting, emphasizing its potential for directly manipulating gradients to enhance resistance to noise.

**Group B in SAM primarily mitigates noisy fitting.** By design, SAM reduces the magnitudes of gradient components in Group B compared to SGD, thereby inhibiting their movement toward local minima. This mechanism is intended to decelerate the convergence of these parameters, prompting the critical question: *Is this down-weighting **mainly** responsible for the observed reduction in noisy fitting in SAM?* To address this question, we conduct experiments aimed at identifying which group of components predominantly contributes to resistance against noisy fitting.

In particular, we compared SAM with a SAM-variant $g^{\text{SAM'}}$ by replacing value of the Group B's gradient components with value of the SGD gradient components while retaining SAM's gradients for the other parameters as follows:

$$g_i^{\text{SAM'}} = \begin{cases} g_i^{\text{SGD}} & \text{if } i \in \mathcal{S}_B, \\ g_i^{\text{SAM}} & \text{otherwise.} \end{cases}$$

We trained on ResNet18 with CIFAR-10 with 25% label noise, following the experimental setup outlined in Appendix A.1. As illustrated in Figure 3, when SAM does not decelerate the gradients of Group B (denoted as SGD-GrB in the figure), the noisy accuracy significantly increases, potentially approaching the noisy accuracy of SGD. This emphasizes that Group B contributes significantly to the noisy fitting resistance of SAM.

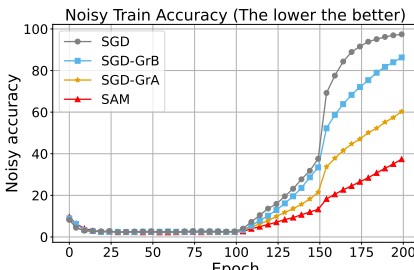

To establish a comparison with Group B, a similar experiment was carried out for Group A. The results from Figure 3 show that, the noisy accuracy does not increase to the same degree as it does for Group B when SGD gradients (shown as SGD-GrA in the figure) replace SAM's up-weighting of gradients in Group A. These findings suggest that Group B play a more significant role in mitigating noisy fitting than Group A.

Figure 3: Comparison of the noisy accuracy of SGD, SAM, and SAM variants where gradient components from groups A and B are swapped with those from SGD.

## 4 ANALYZING THE IMPACT OF GROUP B MAGNITUDE REDUCTION ON SLOWING DOWN NOISY VS. CLEAN LABEL FITTING

From Section 3, we know that Group B slows down the noisy fitting in SAM. As a natural modification, we aim to further slow down Group B by reducing its magnitude. However, we first need to determine if this reduction has a greater impact on fitting clean labels or noisy labels. In this section, we observe that the ratio of Group B in noise-dominated components is significantly higher than the ratio of Group B in clean-dominated components. To be more specific, these components are defined as follows.

To begin, we represent the total gradient in Stochastic Gradient Descent (SGD) as:

$$g^{\text{SGD}} = g^{\text{clean}} + g^{\text{noise}},$$

where $g^{\text{clean}}$ and $g^{\text{noise}}$ denote the gradients derived from the backward passes of clean and noisy examples within a mini-batch, respectively. $g^{\text{SGD}}$ is the aggregated gradient of both.

We focus on the gradient components $g_i^{\text{SGD}}$, particularly when there is opposing interaction between the clean ($g_i^{\text{clean}}$) and noisy ($g_i^{\text{noise}}$) gradients. Our intuition is that when the gradient components of clean and noisy samples align, it is difficult to determine whether the decrease in group B's value is beneficial for resisting noisy fitting. Conversely, when these components oppose each other, a decrease in value will either slow down noisy fitting or slow down clean fitting. Therefore, we define the set of indices where this opposition occurs as:

$$\mathcal{S}_o = \{i \in \{1, 2, \ldots, d\} \mid g_i^{\text{clean}} \cdot g_i^{\text{noise}} < 0\}.$$

Next, we classify gradient components into two sets: *clean-dominated components* $\mathcal{S}_c$ and *noise-dominated components* $\mathcal{S}_n$, based on the predominant influence, as follows:

$$\mathcal{S}_c = \{i \in \{1, 2, \ldots, d\} \mid \underbrace{g_i^{\text{clean}} \cdot g_i^{\text{SGD}} > 0}_{g_i^{\text{SGD}} \text{ is dominated by clean}} \} \cap \mathcal{S}_o; \tag{5}$$

$$\mathcal{S}_n = \{i \in \{1, 2, \ldots, d\} \mid \underbrace{g_i^{\text{noise}} \cdot g_i^{\text{SGD}} > 0}_{g_i^{\text{SGD}} \text{ is dominated by noise}} \} \cap \mathcal{S}_o. \tag{6}$$

To evaluate the influence of clean-dominated and noise-dominated components within group B, we compute their proportions relative to the total number of components. Such an approach is necessary to counterbalance the disparity between number of clean-dominated and noise-dominated components in the gradient. Let $p_{\text{noise}}$ and $p_{\text{clean}}$ denote the proportions of noise-dominated and clean-dominated components, respectively:

$$p_{\text{clean}} = \frac{|\mathcal{S}_c \cap \mathcal{S}_B|}{|\mathcal{S}_c|}; \quad p_{\text{noise}} = \frac{|\mathcal{S}_n \cap \mathcal{S}_B|}{|\mathcal{S}_n|}.$$

We then compute the ratio $pr$, which compares the prevalence of noise-dominated components relative to clean-dominated components:

$$pr = \frac{p_{\text{noise}}}{p_{\text{clean}}}.$$

The ratio $pr$ offers insight into how group B influences the learning process. Specifically, $pr > 1$ indicates that adjustments to the group primarily affect noisy data fitting, while $pr < 1$ suggests a greater influence on clean fitting. Furthermore, a higher value of $pr$ indicates a stronger effect of adjusting Group B on the noisy fitting rate.

**Group B shows stronger impact on noisy fitting.** Figure 4 illustrates the $pr$ values for Group B, using ResNet18 trained on CIFAR-10 with 25% label noise, as discussed in Section 3. The $pr$ ratio in Group B is less than 1 in the early stages of training. However, after 25 epochs, Group B consistently shows $pr > 1$, increasing to values as high as 2 as the neural network begins to overfit noisy labels in the later training phases. This finding reconfirms the impact of Group B on noisy fitting, aligning with the insights presented in Section 3, where replacing Group B with SGD values leads to a dramatic increase in noisy fitting due to the rising of $pr$ value.

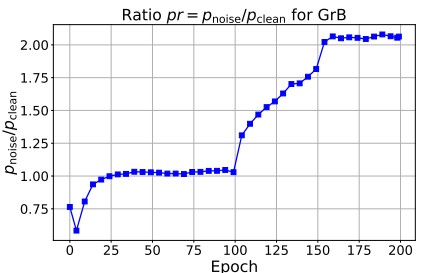

Figure 4: $pr$ value during training, showing that Group B has a greater influence on the noisy fitting rate.

Furthermore, it demonstrates that Group B exerts a stronger influence on noisy data fitting than on clean data fitting as the noisy fitting rate increases. An appropriate reduction in Group B can potentially slow down noisy fitting without hindering the ability to learn from clean samples, which motivates our proposed method. The values of $p_{\text{noise}}$ and $p_{\text{clean}}$ during each iteration and further analyses of Group B are detailed in Appendix B.3.1.

## 5 REWEIGHTING GROUP B FOR ENHANCING NOISY LABEL FITTING RESISTANCE

In this section, we present **SANER** (**S**harpness-**A**ware **N**oise-**E**xplicit **R**eweighting), a novel approach to address noisy fitting that builds on the insights from Sections 3 and 4. Our method demonstrates superior generalization performance compared to SAM and various SAM-based optimizers in noisy label environments across various datasets. We further investigate SANER's effectiveness in mitigating overfitting in three challenging scenarios, following Nakkiran et al. (2020): increasing width of model layer, training without data augmentation, and limited dataset sizes.

### 5.1 SHARPNESS-AWARE NOISE-EXPLICIT REWEIGHTING

Based on finding in Section 3, our proposed method, SANER, aims to enhance SAM's ability to slow down the fitting of noisy labels by straightforwardly reweighting Group B. To achieve this, we first compute a binary mask, $\boldsymbol{m}_{\text{B}}$, which is used to selectively update the gradients as follows:

$$\boldsymbol{m}_{\text{B}} = \begin{cases} 1 & \text{if } 0 \leq r_i < 1, \\ 0 & \text{otherwise}; \end{cases} \tag{7}$$

$$\boldsymbol{g}^{\text{SANER}} = (1 - \boldsymbol{m}_{\text{B}}) \cdot \boldsymbol{g}_{\text{SAM}} + \alpha \cdot \boldsymbol{m}_{\text{B}} \cdot \boldsymbol{g}_{\text{SAM}}. \tag{8}$$

Here, the ratio $\boldsymbol{r}$ is calculated as $\boldsymbol{g}^{\text{SAM}}/\boldsymbol{g}^{\text{SGD}}$ using component-wise operator. It is important to note that SANER maintains the computational efficiency of SAM, as it does not require additional gradient calculations. The complete procedure for SANER is described in Algorithm 1.

---

**Algorithm 1** Sharpness-Aware Noise-Explicit Reweighting (SANER)

---

1: **Input:** Learning rate $\eta$, initial parameters $\boldsymbol{w}_0$, number of iterations $T$, perturbation size $\rho$, noise control parameter $\alpha$
2: Initialize model parameters: $\boldsymbol{w} = \boldsymbol{w}_0$
3: **for** $t = 0$ to $T$ **do**
4:      Sample a mini-batch of $m$ training examples to calculate gradient: $\{\boldsymbol{x}^{(1)}, \ldots, \boldsymbol{x}^{(m)}\}$
5:      Compute the SGD gradient: $\boldsymbol{g}^{\mathrm{SGD}} = \nabla_{\boldsymbol{w}} L(\boldsymbol{w})$
6:      Compute the SAM gradient: $\boldsymbol{g}^{\mathrm{SAM}} = \nabla_{\boldsymbol{w}} L(\boldsymbol{w})\Big|_{\boldsymbol{w}+\rho\frac{\boldsymbol{g}^{\mathrm{SGD}}}{||\boldsymbol{g}^{\mathrm{SGD}}||}}$
7:      Calculate the gradient ratio: $\boldsymbol{r} = \boldsymbol{g}^{\mathrm{SAM}}/\boldsymbol{g}^{\mathrm{SGD}}$ (Component-wise operator)
8:      Compute $\boldsymbol{m}_{\mathrm{B}}$ and $\boldsymbol{g}^{\mathrm{SANER}}$ by Eq. 7 and 8
9:      Update parameters: $\boldsymbol{w} = \boldsymbol{w} - \eta\boldsymbol{g}^{\mathrm{SANER}}$
10: **end for**
11: **Output:** Final learned parameters $\boldsymbol{w}$

---

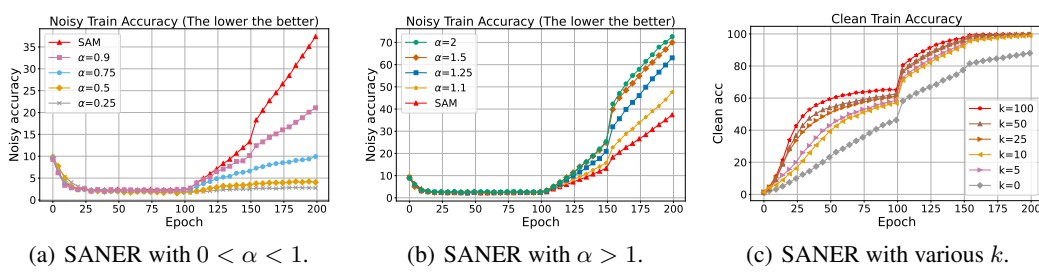

(a) SANER with $0 < \alpha < 1$.      (b) SANER with $\alpha > 1$.      (c) SANER with various $k$.

Figure 5: Effect of hyperparameter $\alpha$ on noisy accuracy in (a) and (b). Lower values of $\alpha$ enhance noise resistance. In (c), compare clean accuracy of SANER with and without the $\alpha$ scheduler, demonstrating that the scheduler improves clean training accuracy.

**Value of $\alpha$ is directly proportional to noisy fitting rate.** Based on the $pr$ value of Group B shown in Section 4, we hypothesize that Group B mitigates noisy fitting by down-weighting gradient components associated with noise-dominated parameters, suggesting that further reducing these components could enhance resistance to noisy fitting. To verify this hypothesis, we conduct experiments using SANER with varying $\alpha$ values $\{2, 1.5, 1.25, 1.1, 1\,(\mathrm{SAM}), 0.9, 0.75, 0.5, 0.25\}$ on ResNet18 and CIFAR-10, employing the hyperparameter settings detailed in Appendix A.1. The results, shown in Figure 5, demonstrate that $\alpha$ is directly proportional to the noisy fitting rate: higher values of $\alpha$ (e.g., 2, 1.5, 1.25, 1.1) accelerate noisy fitting compared to SAM ($\alpha = 1$). Conversely, using lower values of $\alpha$ reduces the contribution of noise-dominated components, allowing SANER to better resist noisy fitting and achieve improved generalization performance.

**Stabilizing clean fitting via a scheduler for $\alpha$.** As shown in Figure 4, the value of $pr$ is relatively low during the early phase of training. This implies that employing a low $\alpha$ value during this phase could impede the learning of clean examples. This phenomenon arises from the model's tendency to prioritize learning clean samples over noisy ones during the early iterations of training, as observed in previous works (Liu et al., 2020; 2023). As a result, during these early iterations, clean-dominated components make up a significant portion of the overall model, including Group B.

To address this issue and ensure robust learning of clean examples, we propose a simple yet effective solution: a linear scheduler that gradually decreases $\alpha$ from 1 to a predetermined value over $k$ epochs. This strategy stabilizes the fitting of clean examples in the early training stages, resulting in more consistent performance compared to training without a scheduler. As shown in Figure 5(c), experiments on ResNet34 with CIFAR-100 under 50% label noise demonstrate that SANER without the $\alpha$ scheduler ($k = 0$) significantly harms clean accuracy, reducing it by approximately 15%. Increasing $k$ improves clean accuracy in the initial phase, and the clean fitting rate remains stable towards the end of training across various value of $k$, indicating that the performance is not overly sensitive to $k$. This approach is particularly important in scenarios with high proportions of noisy samples or slow clean fitting rate, as shown in Tables 7 and 8 in Appendix C.5.

Table 1: Test accuracy comparison of SAM and SANER across different noise types and rates, trained on CIFAR-10 and CIFAR-100 with ResNet18. Bold values highlight the highest test accuracy for each noise type and rate.

| Type | Noise | CIFAR-10 | | CIFAR-100 | |
|------|-------|----------|-----|-----------|-----|
| | | SAM | SANER | SAM | SANER |
| Symm. | 25% | $93.05_{\pm 0.17}$ | $\mathbf{94.08}_{\pm 0.11}$ (↑ 1.03) | $69.68_{\pm 0.07}$ | $\mathbf{72.90}_{\pm 0.21}$ (↑ 3.22) |
| | 50% | $88.82_{\pm 0.08}$ | $\mathbf{90.60}_{\pm 0.36}$ (↑ 1.78) | $61.17_{\pm 0.14}$ | $\mathbf{66.34}_{\pm 0.11}$ (↑ 5.17) |
| Asym. | 25% | $94.75_{\pm 0.28}$ | $\mathbf{94.83}_{\pm 0.14}$ (↑ 0.08) | $71.57_{\pm 0.30}$ | $\mathbf{74.64}_{\pm 0.13}$ (↑ 3.07) |
| | 50% | $81.94_{\pm 0.71}$ | $\mathbf{82.25}_{\pm 1.43}$ (↑ 0.31) | $39.11_{\pm 0.50}$ | $\mathbf{40.05}_{\pm 0.51}$ (↑ 0.94) |
| Depen. | 25% | $92.84_{\pm 0.18}$ | $\mathbf{93.67}_{\pm 0.30}$ (↑ 0.83) | $69.46_{\pm 0.24}$ | $\mathbf{72.93}_{\pm 0.29}$ (↑ 3.47) |
| | 50% | $87.32_{\pm 1.17}$ | $\mathbf{90.01}_{\pm 0.62}$ (↑ 2.69) | $58.71_{\pm 0.69}$ | $\mathbf{66.72}_{\pm 0.75}$ (↑ 8.01) |
| Real | - | $86.33_{\pm 0.07}$ | $\mathbf{87.89}_{\pm 0.12}$ (↑ 1.56) | $62.74_{\pm 0.59}$ | $\mathbf{64.75}_{\pm 0.30}$ (↑ 2.01) |

Table 2: Test accuracy comparison of different architectures using SGD, SAM, and SANER on CIFAR-100 (Symmetric noise). Bold values indicate the highest test accuracy for each architecture and noise level.

| Architecture | Param | Noise | SGD | SAM | SANER |
|--------------|-------|-------|-----|-----|-------|
| ResNet34 | 21.3M | 25% | $69.07_{\pm 0.53}$ | $71.10_{\pm 0.83}$ (↑ 2.03) | $\mathbf{74.02}_{\pm 0.22}$ (↑ 2.92) |
| | | 50% | $59.73_{\pm 1.26}$ | $62.49_{\pm 1.18}$ (↑ 2.76) | $\mathbf{67.26}_{\pm 0.28}$ (↑ 4.77) |
| DenseNet121 | 7.0M | 25% | $69.13_{\pm 0.48}$ | $71.61_{\pm 0.49}$ (↑ 2.48) | $\mathbf{73.89}_{\pm 0.64}$ (↑ 2.28) |
| | | 50% | $58.19_{\pm 1.20}$ | $60.74_{\pm 0.72}$ (↑ 2.55) | $\mathbf{64.26}_{\pm 0.62}$ (↑ 3.52) |
| WideResNet40-2 | 2.3M | 25% | $67.81_{\pm 0.27}$ | $69.75_{\pm 0.26}$ (↑ 1.94) | $\mathbf{70.35}_{\pm 0.10}$ (↑ 0.60) |
| | | 50% | $60.51_{\pm 0.18}$ | $62.58_{\pm 0.35}$ (↑ 2.07) | $\mathbf{64.71}_{\pm 0.55}$ (↑ 2.13) |
| WideResNet28-10 | 36.5M | 25% | $70.78_{\pm 0.20}$ | $72.56_{\pm 0.18}$ (↑ 1.78) | $\mathbf{76.20}_{\pm 0.41}$ (↑ 3.64) |
| | | 50% | $61.94_{\pm 0.49}$ | $64.12_{\pm 0.30}$ (↑ 2.18) | $\mathbf{70.80}_{\pm 0.28}$ (↑ 6.68) |

## 5.2 SETUP AND EXPERIMENTAL RESULTS

**Dataset.** To evaluate the effectiveness of SANER, we assess its performance on CIFAR-10/100 (Krizhevsky et al., 2009) and Mini-WebVision (Li et al., 2017) datasets. We specifically examine four types of label noise—(1) symmetric noise, (2) asymmetric noise (Zhang & Sabuncu, 2018), (3) instance-dependent noise Xia et al. (2020), and (4) real-world noise—on the CIFAR datasets. Details of each noise type are provided in Appendix A.2.

**CIFAR-10 and CIFAR-100.** We validate that SANER can enhance the noise robustness over SGD and SAM trained on ResNet18 (He et al., 2016), achieving the better test accuracy, shown in Table 1. In particular, SANER can outperform SAM in all cases, about CIFAR-10, the improvement over SAM is about 1% average, and the highest improvement is 2.7% for the case of dependent noise type and 50% label noise. While the gap improvement of CIFAR-100 over SAM is about 3% average, and the highest improvement is impressive about 8% for the case of dependent noise type and 50% label noise. These results highlight that SANER effectively enhances model performance by slowing down noisy fitting, particularly in cases like CIFAR-100, where fitting to clean samples is more challenging. This slowdown extends the gap between clean accuracy and noisy accuracy, contributing to better generalization.

**Different architectures.** The performance of optimizers can be highly dependent on the neural network architecture, as different architectures have unique characteristics in terms of depth, width, and connectivity. By evaluating SGD, SAM, and SANER on CIFAR-100 across architectures like ResNet34 (He et al., 2016), DenseNet121 (Huang et al., 2017), WideResNet40-2 and WideResNet28-10 (Zagoruyko & Komodakis, 2017), we aim to compare their test accuracy and assess how well SANER adapts to diverse architectures. The results illustrated in Table 2, SANER consistently outperforms SAM across all tested architectures, with improvements mostly ranging from 2% to over 6%. This significant enhancement provides strong evidence of SANER's robustness across various network designs.

Table 3: Test accuracy comparison of different SAM-like optimizers with and without SANER integration on ResNet18 and CIFAR-10/CIFAR-100 (Symmetric noise). Bold values indicate the highest test accuracy for each optimizer and noise level.

| Optimizer | Noise | CIFAR-10 | | CIFAR-100 | |
|---|---|---|---|---|---|
| | | Original | +SANER | Original | +SANER |
| ASAM | 25% | $92.88_{\pm0.13}$ | $\mathbf{92.96}_{\pm0.06}$ (↑ 0.08) | $70.67_{\pm0.40}$ | $\mathbf{72.44}_{\pm0.10}$ (↑ 1.77) |
| | 50% | $88.70_{\pm0.18}$ | $\mathbf{88.80}_{\pm0.10}$ (↑ 0.10) | $63.04_{\pm0.25}$ | $\mathbf{66.62}_{\pm0.13}$ (↑ 3.58) |
| GSAM | 25% | $93.10_{\pm0.12}$ | $\mathbf{94.09}_{\pm0.16}$ (↑ 0.99) | $69.65_{\pm0.39}$ | $\mathbf{72.97}_{\pm0.27}$ (↑ 3.32) |
| | 50% | $88.71_{\pm0.15}$ | $\mathbf{90.69}_{\pm0.17}$ (↑ 1.98) | $61.25_{\pm0.33}$ | $\mathbf{66.19}_{\pm0.15}$ (↑ 4.94) |
| FSAM | 25% | $92.93_{\pm0.08}$ | $\mathbf{94.00}_{\pm0.15}$ (↑ 1.07) | $69.49_{\pm0.35}$ | $\mathbf{72.94}_{\pm0.58}$ (↑ 3.45) |
| | 50% | $88.71_{\pm0.13}$ | $\mathbf{90.47}_{\pm0.01}$ (↑ 1.76) | $61.24_{\pm0.32}$ | $\mathbf{66.25}_{\pm0.15}$ (↑ 5.01) |
| VaSSO | 25% | $92.35_{\pm0.12}$ | $\mathbf{93.31}_{\pm0.32}$ (↑ 0.96) | $68.86_{\pm0.18}$ | $\mathbf{72.43}_{\pm0.46}$ (↑ 3.57) |
| | 50% | $87.93_{\pm0.06}$ | $\mathbf{89.66}_{\pm0.57}$ (↑ 1.73) | $60.46_{\pm0.05}$ | $\mathbf{65.55}_{\pm0.51}$ (↑ 5.09) |

Table 4: Top-1 validation accuracy (%) on the clean ImageNet 2012 validation set for ResNet18 models trained on WebVision under the Mini setting. Bold values indicate the highest performance for each architecture.

| Architecture | Param | SGD | SAM | SANER |
|---|---|---|---|---|
| ResNet18 | 11.2M | 64.96 | 67.48 | **70.84** |

**Integration with SAM-based optimizers.** We evaluated SANER's effect on SAM-based optimizers using CIFAR-10 and CIFAR-100 with ResNet18. The SAM variants tested include ASAM (Kwon et al., 2021), GSAM (Zhuang et al., 2022), FSAM (Li et al., 2024), and VaSSO (Li & Giannakis, 2024), with and without SANER integration. Table 3 shows that integrating SANER consistently enhances test performance across all noise levels. In challenging scenarios, such as limited samples per class (CIFAR-100) and a high noise rate (50%), SANER integration significantly improves accuracy by around 4-5%. This result suggests that not only does SAM benefit from SANER, but its variants also exhibit similar characteristics related to Group B, which contribute to the mitigation of noisy fitting, as further detailed in Appendix E.1.

**Mini WebVision.** To evaluate SANER beyond the CIFAR benchmarks, we tested it on the large-scale, real-world noisy dataset WebVision (Li et al., 2017). Following the "Mini" setting from previous works (Jiang et al., 2018), we used the first 50 classes from the Google resized image subset and evaluated the networks on the corresponding 50 classes of the clean ImageNet 2012 validation set (Russakovsky et al., 2015). We set up the experiment based on Wei et al. (2023). As shown in Table 4, SANER outperforms SAM by 3% in test accuracy, demonstrating its effectiveness in enhancing noise robustness in large-scale, real-world datasets.

### 5.2.1 EXPERIMENTAL RESULTS IN OVERFITTING SCENARIOS

We further examine SANER's performance compared to SGD and SAM in overfitting environments, by considering three challenging scenarios, following Nakkiran et al. (2020): increasing width of model layers, training without data augmentation, and limited dataset sizes.

**Increasing width of model layers.** We investigate the performance of SANER in an overparameterized regime by increasing the layer widths of the ResNet18 model trained on CIFAR-100. Expanding the model capacity typically accelerates overfitting and memorization of noisy examples (Belkin et al., 2019; Nakkiran et al., 2020; Zhang et al., 2021), leading to degraded performance. This effect is evident from the decline in test accuracy for both SGD and SAM as the ResNet layer width increases, especially with a 50% noise rate (Figure 6(a)). In contrast, SANER's performance improves with increased layer width, leveraging the added capacity more effectively. The performance gap between SANER and SAM grows significantly, reaching approximately 15% in test accuracy when the ResNet18 width is doubled under 50% label noise. These results highlight SANER's robustness in overparameterized settings.

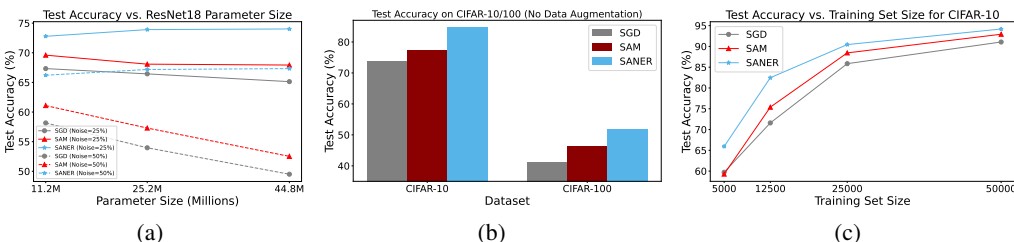

(a)          (b)          (c)

Figure 6: Test accuracy comparison of ResNet18 under different conditions and noise levels: (a) increasing layer width with 25% and 50% label noise, (b) no data augmentation with 25% label noise, and (c) varying CIFAR-10 training set size. SANER consistently outperforms other methods across all settings.

**Without data augmentation.** To further investigate the impact of overfitting, particularly in the presence of noisy examples, we conduct a comparative analysis of the test accuracy of SGD, SAM, and SANER on the CIFAR-10 and CIFAR-100 datasets using ResNet18, under conditions without data augmentation. Data augmentation is often employed to improve model generalization, but its absence allows us to better understand the intrinsic behavior of optimizers when training on raw data. The results, depicted in Figure 6(b), show that SANER achieves a substantial improvement in test accuracy, outperforming both SGD and SAM by a significant margin—most notably, a 7.5% increase on CIFAR-10 compared to SAM.

**Limited dataset sizes.** Finally, a key factor in overfitting is the relationship between the size of the training set and the complexity of the model, which becomes especially pronounced when the dataset is reduced in size. By varying the training set size to 10%, 25%, and 50% of the original data, we aim to examine how SGD, SAM, and SANER respond to reduced data availability on the CIFAR-10 dataset using ResNet18, while maintaining the full-size test set for evaluation. We only test on CIFAR-10 because the number of samples per class in CIFAR-100 is already low, which can lead to unstable training with further data reduction. The results, depicted in Figure 6(c), show that SANER achieves a substantial improvement in test accuracy, outperforming both SGD and SAM by a significant margin—most notably, a 7% increase on CIFAR-10 compared to SAM when the training set is reduced by four times (12,500 examples).

## 6 CONCLUSION

In this work, we examine the effectiveness of the SAM optimizer in addressing label noise, noting its advantages over SGD. However, SAM tends to overfit to noisy labels in later training stages. Our analysis identifies specific down-weighted gradient magnitude components in the SAM's gradient vector that enhance its resistance to label noise. We introduce the concepts of noise-dominated and clean-dominated components, and analyze the impact of Group B on both noisy and clean fitting. Our findings suggest that reducing the magnitude of these down-weighted components can further improve resistance to noisy labels. To this end, we propose SANER, a method designed to reduce these components, resulting in enhanced model robustness. SANER outperforms SAM across different architectures, datasets, and noise scenarios, and demonstrates superior performance compared to other SAM variants when combined with them.

Our work utilizes SAM's perturbed gradient and analyzes it through component-wise gradients to understand their properties in the context of noisy label tasks. However, our design of Group B does not fully separate noise-dominated components from clean-dominated ones, limiting our ability to specifically target noisy label challenges. This overlap may inadvertently affect the clean fitting rate, a drawback that is only partially mitigated by the scheduler we introduced. Additionally, we were limited by the computational resources required to train on larger datasets, such as ImageNet, which could further validate the scalability of our approach. Future research could explore better isolation of noise-dominated components, advanced gradient techniques, or new ways of decomposing SAM's components to enhance label-noise robustness while preserving clean fitting performance.

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

## A  Implementation Details

### A.1  Training Details

We train all neural networks from scratch using simple data augmentation techniques, including `RandomHorizontalFlip(.)` and `RandomCrop(.)`. Specifically, we train the network for 200 epochs using SGD with a momentum of 0.9, a weight decay of 0.0005, and a batch size of 128. The initial learning rate is set to 0.1, and it is reduced by a factor of 10 after 100 and 150 epochs, as suggested by Andriushchenko & Flammarion (2022); Shin et al. (2023). For the hyperparameter perturbation radius of SAM, we use $\rho = 0.1$ in all experiments, following the setting in Foret et al. (2021). For our SANER method, we experiment with different values for the hyperparameter $\alpha$ in the set $\{0.9, 0.75, 0.5, 0.25, 0.1\}$ and found that $\alpha = 0.5$ is stable in most cases, making it our recommended default value. As discussed in Section 5.1, we linearly reduced $\alpha$ from 1 to 0.5 over the first quarter of the total epochs and maintained $\alpha$ at 0.5 for the remaining epochs. In each experiment, we train a neural network on the training dataset and report the best test accuracy on the test dataset during each epoch. We repeat the experiments three times with different random seeds and report the mean and empirical standard deviation of these best results.

### A.2  Types of Noise

In this work, we follows the experimental setup used in previous works for noisy label scenarios. In particular, we used four types of noise as follows:

1. Symmetric noise: Each label is flipped to any other class with equal probability. In our experiments, we uniformly flip labels to other classes with probabilities of 25% and 50%.

2. Asymmetric noise: Labels are flipped to similar, but not identical classes (Zhang & Sabuncu, 2018). For CIFAR-10, we generate asymmetric noisy labels by mapping specific classes to their most similar counterparts: TRUCK to AUTOMOBILE, BIRD to AIRPLANE, DEER to HORSE, CAT to DOG, and leaving other labels unchanged, with probabilities of 25% or 50%. For CIFAR-100, each class is shifted circularly to the next class with probabilities of 25% or 50%.

3. Instance-dependent noise: The mislabeling probability of each instance depends on its input features. In our experiments, we use instance-dependent noise from PDN (Xia et al., 2020) with noisy rates of 25% or 50%, where the noise is synthesized based on DNN prediction errors.

4. Real-world noise: Labels are taken from the mislabeling of real-world human annotations. For CIFAR datasets, we use the "Worst" label set from CIFAR-10N and the "Fine" label set from CIFAR-100N (Wei et al., 2022).

## B  Gradient Behavior in SAM Across Architectures

### B.1  Distribution of Groups A, B, and C

In Section 3, we visualized the distribution of Groups A, B, and C, demonstrating that SAM gradients not only include upweighted gradients, as noted in Baek et al. (2024), but also contain a significant proportion of downweighted gradients. In this section, we extend the analysis by first visualizing these distributions across different $\rho$ values of SAM using ResNet-18 on CIFAR-10, as shown in Figure 7. Next, we analyze the distributions across various architectures (ResNet-34, WideResNet40-2, and DenseNet121) on CIFAR-100, presented in Figure 8. These results further support our findings as outlined in Section 3.

### B.2  Role of Group B in Mitigating Noisy Fitting

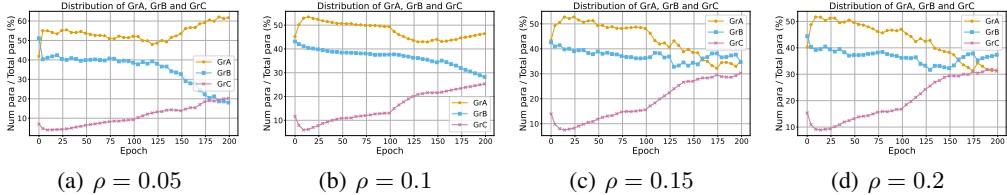

Figure 7: Parameter distribution (%) of groups A, B, and C with different $\rho$ trained on CIFAR-10, 25% label noise.

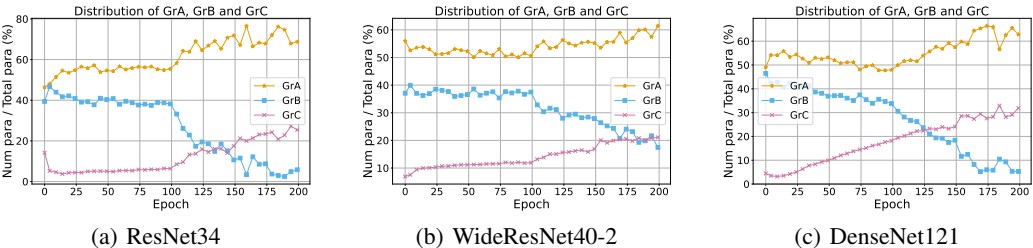

Figure 8: Parameter distribution (%) of groups A, B, and C with different architectures trained on CIFAR100, 25% label noise.

In Section 3, we also conducted experiments where the gradients in each group were replaced with SGD gradients to analyze the contribution of each group in mitigating noisy fitting. In this section, we extend these experiments to various datasets and noise ratios to further validate our finding that Group B plays a primary role in this phenomenon.

As shown in Figure 9, in most cases, the performance of SGD-GrB in resisting noisy label fitting is worse or comparable to SGD-GrA (see the notation in Section 3). An exception is observed with DenseNet121, where SGD-GrA performs worse than SGD-GrB in resisting noise. However, during the later training phase, the model is already overfitted to noise, which slightly impacts the overall performance, as the model's peak performance often occurs between epochs 100 and 150.

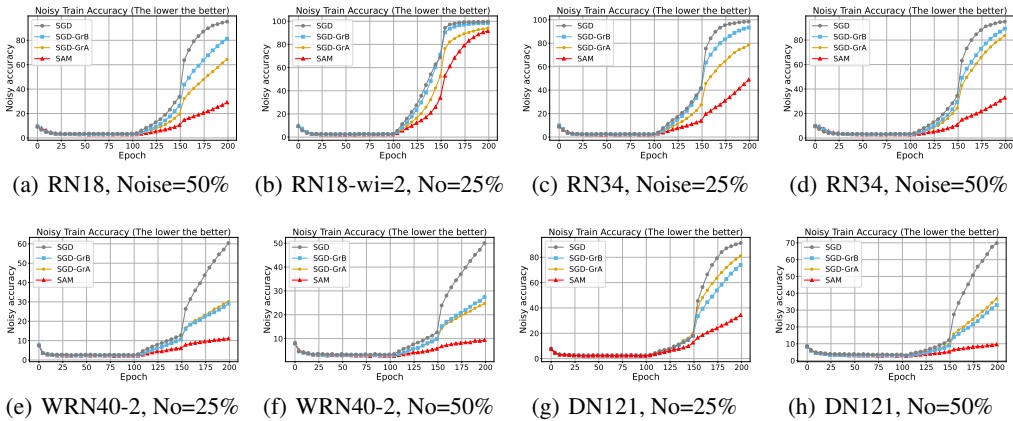

Figure 9: Comparison of noise accuracy: SGD, SAM, SGD-GrA, and SGD-GrB. The noise accuracy of SGD-GrB is higher than that of SAM and is higher or nearly equal to that of SGD-GrA in most cases. This indicates that Group B significantly contributes to the label noise resistance of SAM. These experiments are trained on CIFAR-10.

## B.3 $pr$-VALUES ACROSS ARCHITECTURES

In Section 4, we introduced the $pr$ value to analyze the impact of Group B on noise fitting and clean fitting. A $pr$ value greater than 1 indicates that Group B has a stronger influence on noise fitting. We demonstrated that modifying Group B significantly affects the noise fitting rate, with $pr$ values reaching as high as 2.

To further observe this characteristic of Group B, we conducted experiments using various architectures trained on CIFAR-10 and CIFAR-100 with 25% label noise. As shown in Figure 10, the $pr$ value consistently exceeds 1 across different architectures and rises to around 2 as the noisy fitting rate increases, particularly after epoch 100.

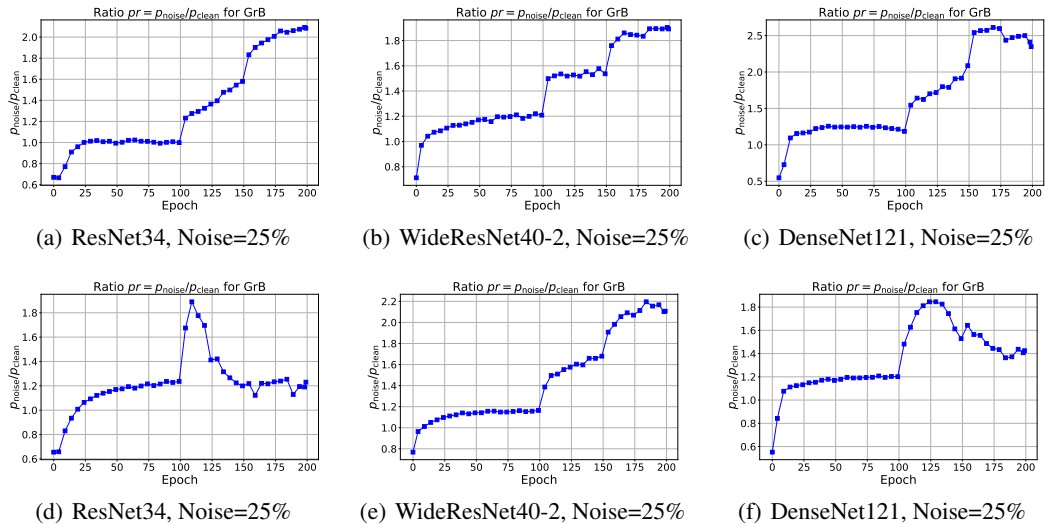

(a) ResNet34, Noise=25%  (b) WideResNet40-2, Noise=25%  (c) DenseNet121, Noise=25%

(d) ResNet34, Noise=25%  (e) WideResNet40-2, Noise=25%  (f) DenseNet121, Noise=25%

Figure 10: $pr$ value during training, showing that Group B has a greater influence on fitting noisy labels compared to clean labels. Experiments in Figures (a)-(f) are trained on CIFAR-10, experiments in Figures (g)-(i) are trained on CIFAR-100.

### B.3.1 ADDITIONAL ANALYSES FOR SECTION 4

To further analyze the value of $pr$ in SANER, we compare the $pr$ during the training process of both SANER and SAM to gain a deeper understanding of SANER's effects on $pr$. Figures 11(a) and 12(a) show that, during SAM training, the $p_{\text{noise}}$ value for group B exceeds $p_{\text{clean}}$ during the noisy label fitting phase. For CIFAR-10, as illustrated in Figure 11(b), SAM effectively mitigates noisy fitting, resulting in a slower increase in $p_r$ compared to CIFAR-100 (Figure 12(b)). Notably, in CIFAR-100, the $pr$ value peaks much earlier (around the 115th epoch) compared to CIFAR-10 (around the 150th epoch), after which it decreases as the model reaches high accuracy on noisy samples, as shown in Figures 11(c) and 12(c). In conclusion, $p_{\text{clean}}$ consistently remains larger than $p_{\text{noise}}$ during the noisy fitting phase when using SAM, and the rate of increase in $pr$ indicates a faster overfitting to noisy samples.

## C HYPERPARAMETER ABLATION STUDY

### C.1 IMPACT OF SAM'S $\rho$ ON SANER PERFORMANCE

Our method builds upon SAM, which has demonstrated the ability to mitigate the effects of label noise. To enhance SAM's generalizability, Foret et al. (2021) suggests increasing the $\rho$ value. In our experiments, we evaluated the effectiveness of SANER across various $\rho$ values, as recommended in Foret et al. (2021). The results, summarized in Table 5, show that SANER consistently outperforms SAM across all tested $\rho$ values. Notably, increasing $\rho$ improves the performance of both SAM and

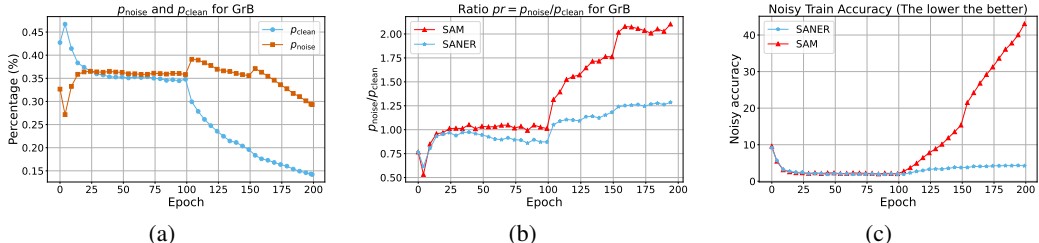

Figure 11: Analysis of Group B trained with ResNet-18 on **CIFAR-10** under 25% label noise. (a) Comparison of $p_{\text{clean}}$ and $p_{\text{noise}}$ using SAM, (b) $pr$ values with SAM versus SANER, and (c) noisy training accuracy. The higher $p_{\text{clean}}$ compared to $p_{\text{noise}}$ helps control group B to mitigate noisy fitting. The $pr$ value increases significantly more slowly for SANER, indicating a gradual increase in the noisy fitting rate, eventually stabilizing at around 5%.

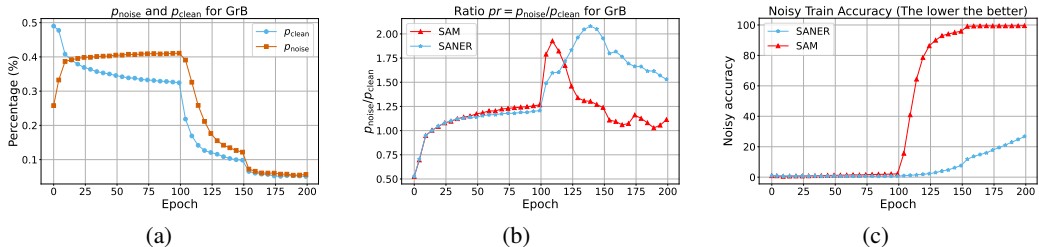

Figure 12: Analysis of Group B trained with ResNet-18 on **CIFAR-100** under 25% label noise. (a) Comparison of $p_{\text{clean}}$ and $p_{\text{noise}}$ using SAM, (b) $pr$ values for SAM versus SANER, and (c) noisy training accuracy. The rapid drop in $p_{\text{clean}}$ is due to the model achieving high noisy training accuracy (nearly 80% by epoch 115). The $pr$ value shows a significant increase with SAM, while it rises more slowly with SANER, correlating with reduced noisy training accuracy.

SANER. However, on CIFAR-100, SANER surpasses SAM's best performance even at the lowest $\rho$ value, with a significant boost in accuracy.

Table 5: Test accuracy comparison of SANER with across different perturbation radii $\rho$ with various noise types and rates, trained on CIFAR-10 and CIFAR-100 with 25% label noise using ResNet18. Bold values highlight the highest test accuracy for each dataset and perturbation radius.

| Dataset | Opt | $\rho = 0.05$ | $\rho = 0.1$ | $\rho = 0.15$ | $\rho = 0.2$ |
|---|---|---|---|---|---|
| CIFAR-10 | SAM | 91.93 | 93.25 | 93.46 | 93.71 |
|  | SANER | **93.32** | **94.18** | **94.09** | **94.23** |
| CIFAR-100 | SAM | 68.33 | 69.60 | 70.30 | 70.86 |
|  | SANER | **72.57** | **72.78** | **73.15** | **73.72** |

## C.2 EFFECT OF $\alpha$ ON PREVENTING NOISE OVERFITTING ACROSS ARCHITECTURES

In Section 5, we demonstrated the correlation between the reweighting factor $\alpha$ and the noisy fitting rate. Specifically, we found that $\alpha$ is directly proportional to the noisy fitting rate, meaning a lower $\alpha$ generally enhances the model's ability to resist noisy fitting.

To further validate this relationship across different architectures, we conducted similar experiments with various architectures, as shown in Figure 13. These results offer insights into how $\alpha$ can be tuned during training to mitigate overfitting.

## C.3 EFFECT OF $k$ IN MITIGATING CLEAN UNDERFITTING ACROSS ARCHITECTURES

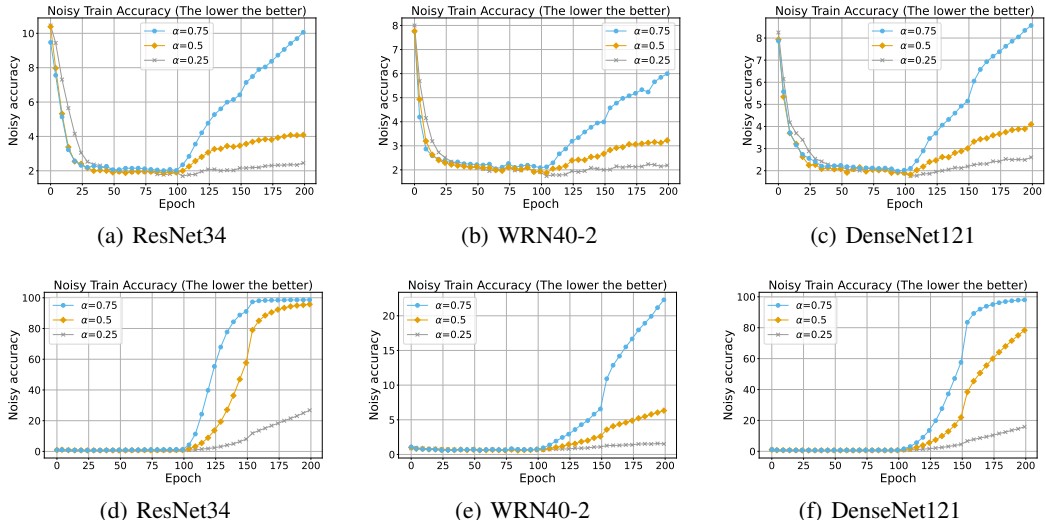

(a) ResNet34      (b) WRN40-2      (c) DenseNet121

(d) ResNet34      (e) WRN40-2      (f) DenseNet121

Figure 13: Noise accuracy, models in Figures (a)-(c) are trained on CIFAR-10 and in Figures (d)-(f) are trained on CIFAR-100 under 25% noisy labels.

In Section 5, we introduced a linear scheduler for the first $k$ epochs of $\alpha$, allowing the model to better fit to clean samples at the start of the training process, given the low $pr$ value at the beginning. This phenomenon is particularly noticeable when the model struggles to fit to clean samples, as seen in CIFAR-100.

To demonstrate the consistency of this characteristic, we visualize the clean training accuracy for different values of $k$ across various architectures in Figure 14. The results show that with $k = 0$ in CIFAR-100, the clean fitting rate of some architectures (e.g., ResNet-34) is slower compared to when $k \geq 25$. Furthermore, in most cases, the $k$ value is not sensitive across a wide range.

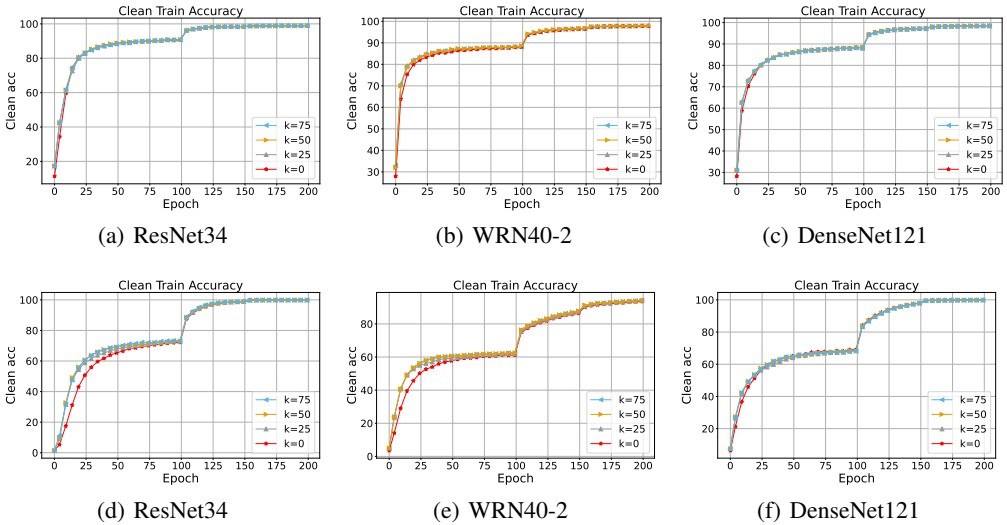

(a) ResNet34      (b) WRN40-2      (c) DenseNet121

(d) ResNet34      (e) WRN40-2      (f) DenseNet121

Figure 14: Clean accuracy, models in Figures (a)-(c) are trained on CIFAR-10 and in Figures (d)-(f) are trained on CIFAR-100 under 25% noisy labels.

## C.4 INTERACTION BETWEEN $\alpha$ AND $k$ IN DIFFERENT SETTINGS

To further explore the interaction between $\alpha$ and $k$ in our proposed methods, we present the grid search results for $\alpha$ and $k$ using ResNet34 and WideResNet40-2 (WRN40-2) on CIFAR-100 with

noisy labels in Table 6. At a low noise rate (25% label noise), models generally avoid underfitting, and $\alpha$ remains robust across a wide range of values (0.25 to 0.75). In this scenario, the scheduler's impact is relatively minor, as the models achieve comparable performance regardless of $k$.

However, under a high noise rate (50% label noise), extremely low values of $\alpha$ (e.g., $\alpha = 0.25$) lead to instability when fitting clean samples, particularly in WRN40-2, where the accuracy drops significantly with small value of $k$ (45.78% at $k = 25$). The scheduler effectively mitigates this instability, as seen in the significant improvement in accuracy with increasing $k$, achieving 55.79% at $k = 75$. Similarly, ResNet34 benefits from the scheduler under high noise conditions, with notable gains in test accuracy. These results demonstrate that while the scheduler has a minor effect under low noise levels, it plays a critical role in stabilizing training and enhancing performance under high-noise conditions as discussed in Section 5, especially when $\alpha$ is small.

Table 6: Test accuracy comparison of SANER trained on CIFAR-100 datasets with noisy labels using ResNet34 and WideResNet40-2 for various $\alpha$ and $k$.

|  |  | Noise = 25% | | | Noise = 50% | | |
| --- | --- | --- | --- | --- | --- | --- | --- |
|  |  | $k = 25$ | $k = 50$ | $k = 75$ | $k = 25$ | $k = 50$ | $k = 75$ |
| | $\alpha = 0.75$ | 73.09 | 72.76 | 72.04 | 66.37 | 66.28 | 64.73 |
| ResNet34 | $\alpha = 0.5$ | 73.96 | 74.03 | 74.01 | 66.93 | 67.17 | 66.73 |
| | $\alpha = 0.25$ | 74.33 | 74.49 | 74.25 | 61.51 | 66.60 | 67.48 |
| | $\alpha = 0.75$ | 70.34 | 70.90 | 70.48 | 63.80 | 64.08 | 63.35 |
| WRN40-2 | $\alpha = 0.5$ | 71.55 | 71.93 | 71.70 | 64.94 | 64.73 | 65.70 |
| | $\alpha = 0.25$ | 71.27 | 71.79 | 71.73 | 45.78 | 52.14 | 55.79 |

## C.5 THE EFFECT OF SCHEDULING DURING TRAINING

To further support our analysis in Appendix B.3.1, we demonstrate the effectiveness of model when $\alpha = 0.5$ without using scheduler in more diverse setup. The results in Tables 7 and 8 reconfirm that while the scheduler can improve performance in certain scenarios, its effectiveness varies depending on the noise level and model architecture. Notably, SANER consistently outperforms SAM across all cases, even without the scheduler.

Table 7 compares SANER with and without the scheduler under different noise types and rates on CIFAR-10 and CIFAR-100. For datasets with a limited number of samples per class, such as CIFAR-100, the scheduler consistently improves test accuracy, particularly under high noise levels (e.g., 50% label noise). This is primarily because fitting clean samples is more challenging during the initial training epochs in such noisy environments. However, for lower noise levels (e.g., 25%) or datasets where clean fitting is inherently easier (e.g., CIFAR-10), the performance improvement is marginal.

Table 8 evaluates the scheduler's impact across various architectures. Generally, the scheduler enhances performance under high-noise conditions. For example, with WideResNet40-2 at 50% noise, it increases accuracy by 2.69%. However, in some cases, such as DenseNet121 with 50% noise, the scheduler offers little to no improvement and may even slightly underperform.

Overall, these results suggest that the scheduler is particularly beneficial for stabilizing clean fitting in high-noise environments. Nevertheless, it may not be critical in all scenarios, with its utility being most apparent when training on noisy datasets where fitting clean data poses a significant challenge.

# D    ABLATION STUDY ACROSS DIVERSE SETUPS

## D.1    CLEAN DATASETS AND VARYING NOISE RATES

### D.1.1    NOISE-FREE SCENARIOS

Table 7: Test accuracy comparison of SANER with and without alpha scheduler across different noise types and rates, trained on CIFAR-10 and CIFAR-100 with ResNet18, we report the highest test accuracy in three different seed experiments. Bold values highlight the highest test accuracy for each noise type and rate.

| Type | Noise | CIFAR-10 | | | CIFAR-100 | | |
|------|-------|----------|--------|--------|-----------|--------|--------|
| | | SAM | SANER | | SAM | SANER | |
| | | | $k = 0$ | $k = 50$ | | $k = 0$ | $k = 50$ |
| Symm | 25% | 93.05 | **94.29** | 94.18 | 69.68 | 73.04 | **73.14** |
| | 50% | 88.82 | 90.09 | **90.93** | 61.17 | 64.78 | **66.41** |
| Asym | 25% | 94.75 | **95.11** | 94.95 | 71.57 | 74.33 | **74.75** |
| | 50% | 81.94 | **83.85** | 82.44 | 39.11 | 40.64 | **40.90** |
| Depen | 25% | 92.84 | 93.86 | **93.99** | 69.46 | 73.16 | **73.25** |
| | 50% | 87.32 | 90.41 | **90.58** | 58.71 | 66.72 | **67.22** |
| Real | - | 86.33 | 87.78 | **88.02** | 62.74 | 64.71 | **65.07** |

Table 8: Test accuracy comparison of different architectures using SAM and SANER with and without alpha scheduler, trained on CIFAR-100. Bold values indicate the highest test accuracy for each architecture and noise level.

| Architecture | Param | Noise | SAM | SANER | |
|--------------|-------|-------|-----|-------|------|
| | | | | $k = 0$ | $k = 50$ |
| ResNet34 | 21.3M | 25% | 71.10 | 74.14 | **74.24** |
| | | 50% | 62.49 | 64.85 | **67.57** |
| DenseNet121 | 7.0M | 25% | 71.61 | 73.78 | **74.61** |
| | | 50% | 60.74 | **66.04** | 64.04 |
| WideResNet40-2 | 2.3M | 25% | 69.75 | 70.12 | **70.47** |
| | | 50% | 62.58 | 62.41 | **65.10** |
| WideResNet28-10 | 36.5M | 25% | 72.56 | 76.45 | **76.55** |
| | | 50% | 64.12 | 69.41 | **70.68** |

In this section, we compare SAM and SANER in noise-free scenarios to evaluate SANER's performance without label noise. The hyperparameter settings are consistent with those outlined in Appendix A.1. The results, shown in Table 9, indicate that SANER does not significantly outperform SAM when label noise is absent. This reinforces the idea that SANER is primarily designed to address the challenges associated with noisy labels.

Nevertheless, SANER does not cause any performance degradation compared to SAM in clean environments. This suggests that SANER's mechanism of reducing Group B gradients, which target potentially noisy or harmful updates, does not negatively affect overall model performance when label noise is absent.

Table 9: Test accuracy comparison of SAM and SANER trained on clean CIFAR-10 and CIFAR-100 datasets using ResNet18. Bold values highlight the highest test accuracy for each dataset.

| Dataset | Noise | SAM | SANER |
|---------|-------|-----|-------|
| CIFAR-10 | 0% | $96.04_{\pm 0.04}$ | $\mathbf{96.06}_{\pm 0.12}$ |
| CIFAR-100 | 0% | $79.19_{\pm 0.22}$ | $\mathbf{79.63}_{\pm 0.36}$ |

### D.1.2 SCENARIOS WITH DIFFERENT NOISE LEVELS

Table 10 presents a comparison of the test accuracy between SAM and SANER across different noise levels (20%, 40%, 60%, and 80%) on the CIFAR-10 dataset using ResNet-32, following the hyperparameter settings in Xie et al. (2024). The results indicate that SANER consistently outper-

forms SAM across all noise rates, demonstrating its effectiveness in mitigating the impact of label noise.

Table 10: Test accuracy comparison of SAM and SANER with various noise rates, trained on CIFAR-10 using ResNet32. Bold values highlight the highest test accuracy for each noise rate.

| Dataset | Opt | Noise rate | | | |
|---------|-----|------|------|------|------|
| | | 20% | 40% | 60% | 80% |
| CIFAR-10 | SAM | 91.44 | 88.62 | 84.41 | 48.40 |
| | SANER | **92.00** | **90.41** | **85.08** | **49.27** |

## D.2 Bootstrapping with SAM and SANER

We conducted experiments to compare the test accuracy of SAM and SANER trained on ResNet-18 with CIFAR-10 and CIFAR-100, integrated with hard bootstrapping (Reed et al., 2014), as used in Foret et al. (2021). The overall results, as shown in Table 11, demonstrate that SANER consistently outperformed both SAM and Bootstrap + SAM across all noise levels for both datasets. SANER achieved higher accuracy compared to SAM at both 25% and 50% noise rates, and it also surpassed Bootstrap + SAM, especially when integrated with bootstrapping.

Table 11: Test accuracy comparison of SAM, SANER integrated with and without Bootstrap trained on CIFAR-10 and CIFAR-100 datasets using ResNet18. Bold values highlight the highest test accuracy for each noise type and rate.

| Dataset | Noise | SAM | Bootstrap + SAM | SANER | Bootstrap + SANER |
|---------|-------|-----|-----------------|-------|-------------------|
| CIFAR-10 | 25% | 93.25 | 93.26 | 94.14 | **94.25** |
| | 50% | 88.77 | 89.33 | **90.65** | 90.64 |
| CIFAR-100 | 25% | 69.60 | 71.01 | 72.78 | **73.45** |
| | 50% | 61.06 | 64.21 | **65.73** | 65.57 |

## E  Training process visualization

### E.1  Integration with SAM variants

**Experimental setup.** To evaluate the effect of SANER on SAM-based optimizers, we conducted experiments on CIFAR-10 and CIFAR-100 using ResNet18. The SAM variants used include ASAM (Kwon et al., 2021), GSAM (Zhuang et al., 2022), FSAM (Li et al., 2024), and VaSSO (Li & Giannakis, 2024), tested both with and without SANER integration. The models were trained with label noise levels of 25% and 50%, and the SANER hyperparameter $\alpha = 0.5$, and $k = 50$ for all experiments as we setup when comparing with SAM. All other training configurations were kept consistent for fair comparison between methods.

**Modification of SAM-based optimizers.** SANER was integrated into the SAM-variants by modifying the update rules. Specifically, we replaced $r$ and $g^{\text{SAM}^*}$ as follows:

$$r = \frac{g^{\text{SAM}^*}}{g^{\text{SGD}}}, \tag{9}$$

$$g^{\text{SANER}^*} = (1 - m_\text{B}) \cdot g^{\text{SAM}^*} + \alpha \cdot m_\text{B} \cdot g^{\text{SAM}^*}, \tag{10}$$

where $g^{\text{SAM}^*}$ refers to the gradient of the specific SAM variant and $g^{\text{SANER}^*}$ denotes the modified gradient under SANER integration. The mask $m_\text{B}$ is computed according to Equation 7.

**Noisy accuracy.** As illustrated in Figure 15, the integration of SANER into SAM variants significantly reduces the number of noisy examples that are memorized during training compared to their original variants. This is particularly evident in high-noise scenarios such as 50%, where the noisy

fitting curve rises more gradually in SANER-integrated models compared to their original counterparts. This indicates that SANER helps slow down the memorization of noisy labels, allowing the models to focus more on clean data, which leads to better generalization.

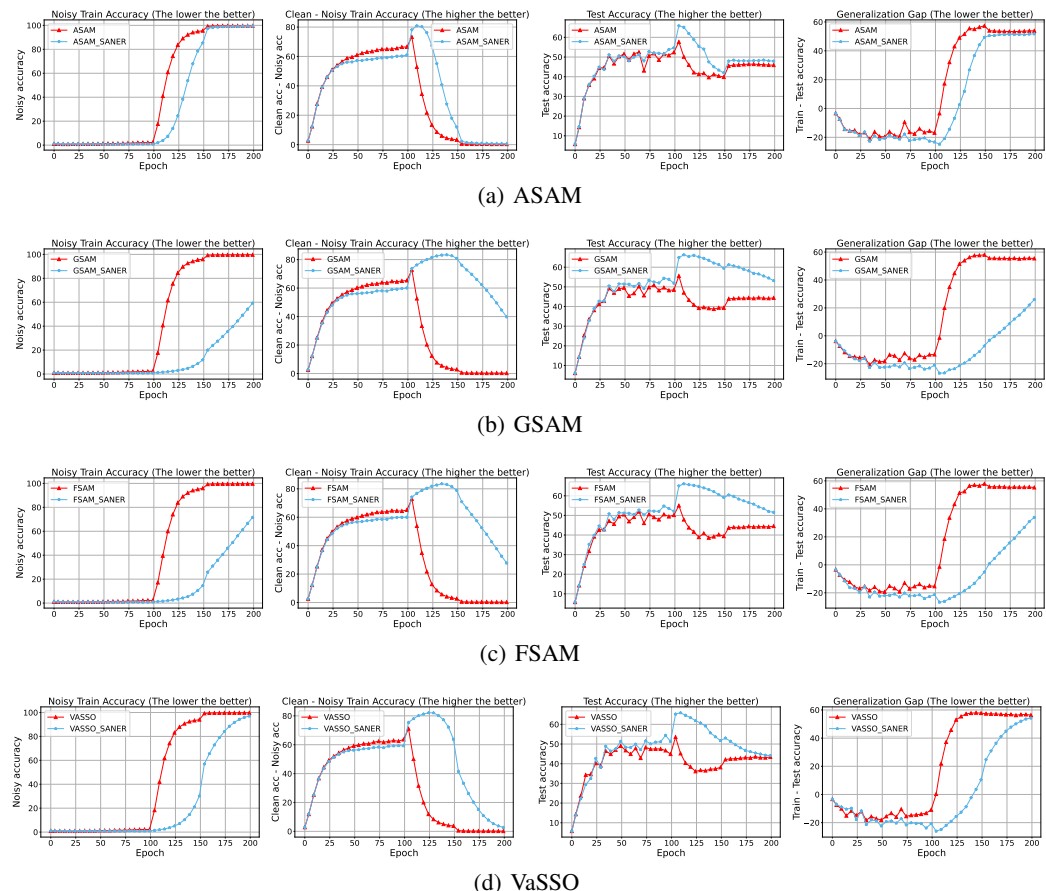

Figure 15: Performance comparison of ASAM, GSAM, FSAM, and VaSSO (with and without SANER) trained on ResNet18 with CIFAR-100 under 25% label noise. The columns show the noisy training accuracy, gap between clean and noisy accuracy, test accuracy, and generalization gap from left to right respectively. Overall, integrating SANER with these SAM variants proves beneficial by slowing the noisy fitting rate while preserving the clean fitting rate.

### E.2 EFFECT OF INCREASING RESNET18 WIDTH

To demonstrate the effectiveness of our method, we conduct experiments in overfitting-prone scenarios by increasing model parameters, as detailed in Section 5.2. In this section, we visualize the training process under overparameterization by increasing the width of ResNet18 to provide further insights into the fitting rates of SGD, SAM, and SANER. As shown in Figure 16, increasing model width enhances overfitting, causing SAM to match the noisy fitting rate of SGD. In contrast, SANER maintains a slower noisy fitting rate while preserving the clean fitting rate, allowing the model to better leverage overparameterization and achieve higher test accuracy.

### E.3 VARIOUS ARCHITECTURES

To evaluate SANER's robustness across different neural network architectures, we conducted experiments using ResNet34, DenseNet121, and WideResNet28-10 on CIFAR datasets with 25% and 50% label noise. We analyze the impact of SANER on the training process, specifically its ability to regulate noisy fitting rates, as shown in Figure 17. SANER consistently achieves better control over noisy fitting, thereby reducing overfitting and enhancing generalization performance. These results

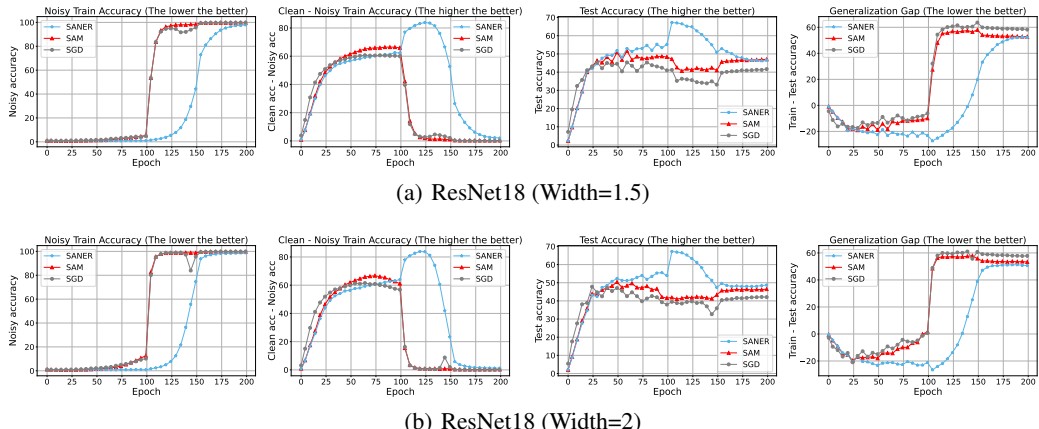

(a) ResNet18 (Width=1.5)

(b) ResNet18 (Width=2)

Figure 16: Performance comparison of SAM, SGD, and SANER (ours) when **increasing width** of ResNet18 trained on CIFAR-100 under 50% label noise. The columns show the noisy training accuracy (1st column), gap between clean and noisy accuracy (2nd column), test accuracy (3rd column), and generalization gap (4th column), respectively. The noisy fitting rate of SAM reaches that of SGD, whereas SANER keeps it low for a longer duration, resulting in better performance.

demonstrate SANER's effectiveness in handling noisy data across diverse architectures, yielding significant improvements over both SGD and SAM.

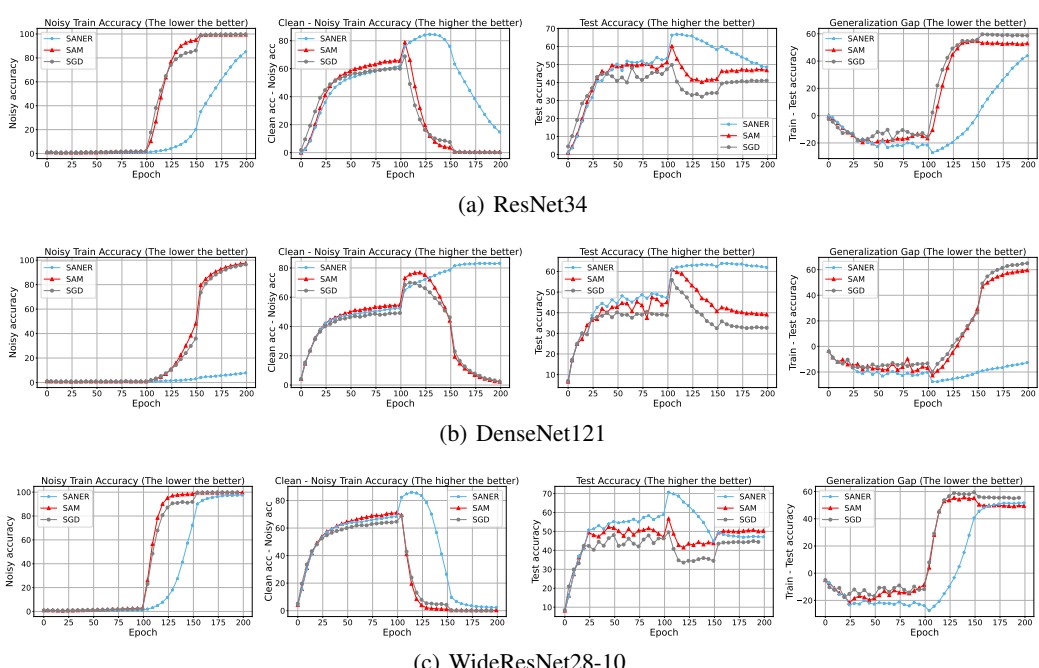

(a) ResNet34

(b) DenseNet121

(c) WideResNet28-10

Figure 17: Performance comparison of SAM, SGD, and SANER (ours) across **different models** trained on CIFAR-100 under 50% label noise. The columns show the noisy training accuracy (1st column), gap between clean and noisy accuracy (2nd column), test accuracy (3rd column), and generalization gap (4th column), respectively. SANER outperforms SAM in both noisy accuracy and the clean-noisy accuracy gap, demonstrating better generalization through higher test accuracy.

