# OpenReview forum: "Improving Resistance to Noisy Label Fitting by Reweighting Gradient in SAM"
_ICLR.cc/2025/Conference — Submitted to ICLR 2025_

### Official Review · Reviewer_xkFP · 2024-10-29

**Soundness:** 3
**Presentation:** 3
**Contribution:** 2
**Rating:** 6
**Confidence:** 4

**Summary:**

This manuscript explores the potential of SAM in the context of noisy label learning. Specifically, it provides an empirical study of the component-wise gradients in SAM, identifying the gradient components most crucial for handling noisy labels. Building on these insights, this manuscript proposes SANER, a method that adjusts SAM gradients to improve model robustness against label noise.

**Strengths:**

1. The writing quality is good.
2. This manuscript provides a detailed experimental analysis of SAM and its robustness to label noise.
3. The proposed method is tested across different model architectures, layer widths, and data sizes, demonstrating its generalization capability across different settings.

**Weaknesses:**

1. The theoretical analysis is lacking. Specifically, there is no justification for why downweighting the gradient component in Group B helps the model resist label noise, or why gradients from noisy samples dominate Group B as the model begins to overfit. A theoretical analysis could suggest a more effective gradient adjustment method than simple downweighting.
2. Some experimental results are missing.
    1. Experimental results for high label noise settings (e.g., 80% symmetric label noise) are absent. Given that this manuscript primarily addresses label noise learning, evaluating the proposed method under high label noise conditions is essential, as it is a common practice in current label noise learning literature.
    2. The clean accuracy results for different values of $\alpha$ are missing. Recall that $p_\text{clean}$ still accounts for 1 / 3 of the total components in Group B, so downweighting Group B may potentially harm the accuracy of clean samples.

**Questions:**

1. Regarding Figure 3, since Group A also contributes to SAM’s resistance to noisy fitting, why not reweight Group A as well?
2. Since SAM enhances model performance in clean label scenarios, would SANER perform worse than SAM in such cases?
3. Are there any improvements when combined with other label noise learning algorithms?

---

> ### Author Response · Authors · 2024-11-21
> **Thanks for your review. We have addressed your concern.**
>
> >*Regarding Figure 3, since Group A also contributes to SAM’s resistance to noisy fitting, why not reweight Group A as well?*
>
> While Group A contributes to SAM's resistance to noisy fitting by upweighting *clean samples* (Baek et al., 2024), this effect *implicitly* addresses the mechanism of slowing down the fitting of noisy samples. Instead, our work focuses on the specific term that *explicitly* mitigates noisy fitting.
>
> To this end, we begin by exploring the gradient behavior of SAM. Our empirical study reveals that Group B inherently slows down the convergence rate and represents a comparable proportion to Group A. This aligns more closely with our objective of addressing noisy fitting *explicitly*, which is analyzed in Section 4. Moreover, as illustrated in Figure 3, Group B exhibits a greater influence on noisy fitting, establishing it as the primary target for intervention.
>
> While reweighting Group A could potentially improve performance, such an analysis requires deeper investigation and is beyond the scope of this work.
>
> >Experimental results for high label noise settings (e.g., 80% symmetric label noise) are absent. Given that this manuscript primarily addresses label noise learning, evaluating the proposed method under high label noise conditions is essential, as it is a common practice in current label noise learning literature.*
>
> We conducted experiments using ResNet-32 and CIFAR-10 with noise levels of 20%, 40%, 60%, and 80%, following Xie et al. (2024). SANER consistently outperformed SAM across various noise ratios, as shown in Table.
>
> | Dataset    | Opt    | Noise = 20% | Noise = 40% | Noise = 60% | Noise = 80% |
> |------------|--------|-------------|-------------|-------------|-------------|
> | **CIFAR-10** | SAM    | 91.44       | 88.62       | 84.41       | 48.40       |
> |             | SANER  | **92.00**   | **90.41**   | **85.08**   | **49.27**   |
>
>
> >*Since SAM enhances model performance in clean label scenarios, would SANER perform worse than SAM in such cases?*
>
> Actually, SANER even has minor performance improvements over SAM in clean label scenarios. We conducted experiments on ResNet18 using the clean CIFAR-10 and CIFAR-100 datasets, as shown in Table. The hyperparameters were configured as described in Appendix A.1 (Training Details) for reference.
>
> | Dataset    | Noise | SAM                | SANER               |
> |------------|-------|--------------------|---------------------|
> | **CIFAR-10** | 0%    | 96.04 ± 0.04       | **96.06 ± 0.12**    |
> | **CIFAR-100**| 0%    | 79.19 ± 0.22       | **79.63 ± 0.36**    |
>
> >*Are there any improvements when combined with other label noise learning algorithms?*
>
> We conducted experiments to compare the test accuracy of SAM and SANER trained on ResNet-18 with CIFAR-10 and CIFAR-100, integrated with hard bootstrapping (Reed et al., 2014), as used in Foret et al. (2021). The results demonstrate that SANER consistently outperformed Bootstrap + SAM, as shown in Table.
>
> | Dataset    | Noise | SAM    | SANER  | Bootstrap + SAM | Bootstrap + SANER |
> |------------|-------|--------|--------|-----------------|-------------------|
> | **CIFAR-10** | 25%   | 93.25  | 94.14  | 93.26           | **94.25**         |
> |             | 50%   | 88.77  | **90.65** | 89.33           | 90.64             |
> | **CIFAR-100**| 25%   | 69.60  | 72.78  | 71.01           | **73.45**         |
> |             | 50%   | 61.06  | **65.73** | 64.21           | 65.57             |
>
> References:
> - Christina Baek, Zico Kolter, and Aditi Raghunathan. Why is sam robust to label noise? ICLR, 2024. URL https://arxiv.org/abs/2405.03676.
> - Pierre Foret, Ariel Kleiner, Hossein Mobahi, and Behnam Neyshabur. Sharpness-aware minimization for efficiently improving generalization. ICLR, 2021.
>
> *Conclusion*
>
> We thank you for your valuable feedback. We believe we have addressed all your concerns and have added experiments related to various noise rates (including noise-free setup) in **Appendix D.1 and D.2**. As a result, we would really appreciate it if you consider increasing your score.

---

> ### Author Response · Authors · 2024-11-23
> **Kind Follow-Up Reminder**
>
> Dear Reviewer,
>
> Thank you once again for reviewing our manuscript. We have done our best to address your comments and have revised the paper based on suggestions from all reviewers.
>
> We understand today is the weekend and do not wish to disturb you. However, with the approaching deadline, we hope you could review the revisions and provide your feedback at your earliest convenience.
>
> Thank you for your valuable insights and support. Please let us know if you have any additional questions or need further clarification.

---

### Official Review · Reviewer_t7xA · 2024-10-31

**Soundness:** 2
**Presentation:** 3
**Contribution:** 2
**Rating:** 5
**Confidence:** 3

**Summary:**

This paper presents SANER, a novel approach designed to improve the robustness of Sharpness-Aware Minimization (SAM) in the presence of noisy labels. The proposed method introduces a reweighting mechanism for gradient components to suppress the fitting to noisy labels, thereby enhancing generalization performance. Extensive experiments are conducted on various datasets, demonstrating the effectiveness of SANER compared to SAM and other optimizers.

**Strengths:**

1）SANER introduces a novel reweighting mechanism for specific gradient components, effectively extending SAM to better handle noisy labels. This innovation is well-motivated and provides a meaningful contribution to the field of noise-resistant optimization.
2）The experimental validation convincingly demonstrates that SANER outperforms SAM and other optimizers in different noise scenarios. The visualizations provided further reinforce the claims regarding the noise robustness of SANER.
3）The use of a linear schedule for adjusting parameters during training is practical and helps stabilize the learning process, especially in the early stages when noisy labels are more problematic.

**Weaknesses:**

1）The paper lacks a detailed explanation of how the gradient components are divided into Group A, B, and C. It is unclear what specific criteria were used for this division, and including pseudocode would enhance clarity and reproducibility. Additionally, while visual evidence supports the reweighting mechanism, systematic mathematical derivations justifying the effectiveness of the approach in suppressing noisy label fitting would be valuable.
2）The manuscript includes comparisons with VaSSO, but there are notable gaps in performance under different noise rates, even with identical experimental settings. Clarifying why these significant differences occur would improve the discussion. Moreover, additional experiments with varied noise ratios would help provide a more balanced comparison. Furthermore, comparing SANER with methods such as AdaSam and Adan could better help demonstrate its broader generalization capabilities or effectiveness across different tasks and domains. AdaSam is primarily used in natural language processing tasks, while Adan is a recent adaptive Nesterov momentum algorithm showing promise in vision, language, and reinforcement learning tasks.
3）The analysis of the sensitivity of the parameter α is limited. A more comprehensive study covering different noise ratios, network architectures, and training conditions would be beneficial. For instance, exploring α’s behavior in noisy segmentation tasks using Vision Transformers (e.g., ViT-based noisy segmentation) as well as standard classification tasks with different architectures (e.g., ResNet versus VGG) would provide a clearer picture of the parameter's impact across varied settings.

**Questions:**

1）Regarding the division of gradient components into Group A, B, and C, what specific criteria or methodology did you use for this grouping? Could you provide pseudocode for this process to make the grouping mechanism clearer? Additionally, could you explain why these particular components were chosen for reweighting, and include systematic mathematical derivations to show how these components effectively suppress noisy label fitting?
2）The paper includes comparisons with VaSSO; however, the results under different noise rates show considerable gaps. Could you explain why such significant differences occur under identical experimental settings? Moreover, could you conduct additional experiments covering more diverse noise ratios to provide a balanced comparison, and clarify whether these differences are due to inherent limitations of the methods or specific experimental setups?
3）In the appendix, only ablation studies for the presence or absence of the α parameter were conducted, but there is a lack of detailed experimental analysis on the selection of α values (e.g., from 0 to 1 or values greater than 1). How do different values of α affect performance across varying noise ratios and network architectures? Providing such detailed experimental support would help in understanding the optimal choice of α under different scenarios.

---

> ### Author Response · Authors · 2024-11-21
> **Thanks for your review. We have addressed your concern. (Part 1/2)**
>
> >*Regarding the division of gradient components into Group A, B, and C, what specific criteria or methodology did you use for this grouping? Could you provide pseudocode for this process to make the grouping mechanism clearer?*
>
> The way to divide these groups is carefully specified in Section 3 and Algorithm 1. We summarize it for you as follows: In the SAM algorithm, we have two gradients, $g^{SAM}$ and $g^{SGD}$. We then calculate the ratio $r = g^{SAM} / g^{SGD}$ (component-wise operator). Finally, we identify the $i$-th parameter as belonging to Group A, Group B, or Group C if $r_i \geq 1$, $0 \leq r_i < 1$, or $r_i < 0$, respectively.
>
> >*Why these particular components were chosen for reweighting, and include systematic mathematical derivations to show how these components effectively suppress noisy label fitting?*
>
> The motivation for selecting Group B for reweighting is detailed in Section 3, with its role in suppressing noisy label fitting further discussed in Section 4. Here is a summary: Baek et al. (2024) have thoroughly discussed how the gradients in Group A *implicitly* prevent noisy fitting. Meanwhile, based on our observations, Group B includes a comparable proportion of parameters as Group A but naturally slows down the convergence rate. This leads us to hypothesize that Group B *explicitly* controls noisy fitting. Hence, our analysis focuses on this group within the SAM gradient vector.
>
> Through our empirical study, we find that SAM loses its ability to resist noisy fitting when the magnitudes in Group B are replaced with those from SGD (detailed in Section 3). Furthermore, our analysis reveals that Group B contains a much larger proportion of "noise-dominated components" (as defined in Section 4) compared to "clean-dominated components." This suggests that reducing the magnitudes of these components can enhance resistance to noisy fitting without significantly affecting the clean fitting rate. As a result, our method consistently outperforms SAM across all cases even with clean data.
>
>
> >*The paper includes comparisons with VaSSO; however, the results under different noise rates show considerable gaps. Could you explain why such significant differences occur under identical experimental settings?*
>
> These differences occur under identical experimental settings, as specified in Appendix A.1 Training Details. The hyperparameters we used for VaSSO are $\rho=0.1$ and $\theta=0.2$, as recommended by Li & Giannakis, (2024). We applied the same hyperparameters when integrating our method, SANER, into VaSSO. The hyperparameters for SANER are $\alpha=0.5$ and $k=50$, which are the default values recommended in our paper.
>
> >*Could you conduct additional experiments covering more diverse noise ratios to provide a balanced comparison, and clarify whether these differences are due to inherent limitations of the methods or specific experimental setups?*
>
> We conducted experiments using ResNet-32 and CIFAR-10 with noise levels of 20%, 40%, 60%, and 80%, following Xie et al. (2024). SANER consistently outperformed SAM across various noise ratios, as shown in Table.
>
> | Dataset    | Opt    | Noise = 20% | Noise = 40% | Noise = 60% | Noise = 80% |
> |------------|--------|-------------|-------------|-------------|-------------|
> | **CIFAR-10** | SAM    | 91.44       | 88.62       | 84.41       | 48.40       |
> |             | SANER  | **92.00**   | **90.41**   | **85.08**   | **49.27**   |
>
>
> >*But there is a lack of detailed experimental analysis on the selection of α values (e.g., from 0 to 1 or values greater than 1)*
>
> We would like to clarify that a detailed experimental analysis of $\alpha$ selection is provided in Section 5. Figures 5a and 5b demonstrate that $\alpha \in (0, 1)$ effectively slows down the fitting of noisy labels, while $\alpha > 1$ accelerates it. Furthermore, we recommend $\alpha=0.5$ as the default value for all our experiments, as specified in Appendix A1: Training Details. Please refer to these sections for more details.

---

> ### Author Response · Authors · 2024-11-21
> **Thanks for your review. We have addressed your concern. (Part 2/2)**
>
> >*How do different values of $\alpha$ affect performance across varying noise ratios and network architectures?*
>
> Based on our analysis of $\alpha$ in Section 5, $\alpha$ is related to the model's fitting ability. If the model shows signs of overfitting, lowering $\alpha$ can help mitigate this issue. However, an extremely low $\alpha$ can lead to underfitting (e.g. 50% label noise). To further support this analysis, we provide a Table evaluating $\alpha \in \{0.25, 0.5, 0.75\}$ and $k \in \{25, 50, 75\}$, trained on ResNet34 and WideResNet40-2 with CIFAR-100 under 25% and 50% noisy labels.
>
> | $\alpha$ and $k$ | Noise = 25% ($k=25$) | Noise = 25% ($k=50$) | Noise = 25% ($k=75$) | Noise = 50% ($k=25$) | Noise = 50% ($k=50$) | Noise = 50% ($k=75$) |
> |-------------------|----------------------|----------------------|----------------------|----------------------|----------------------|----------------------|
> | **RN34**      |                      |                      |                      |                      |                      |                      |
> | $\alpha=0.75$     | 73.09                | 72.76                | 72.04                | 66.37                | 66.28                | 64.73                |
> | $\alpha=0.5$      | 73.96                | 74.03                | 74.01                | 66.93                | 67.17                | 66.73                |
> | $\alpha=0.25$     | 74.33                | 74.49                | 74.25                | 61.51                | 66.60                | 67.48                |
> | **WRN40-2** |                      |                      |                      |                      |                      |                      |
> | $\alpha=0.75$     | 70.34                | 70.90                | 70.48                | 63.80                | 64.08                | 63.35                |
> | $\alpha=0.5$      | 71.55                | 71.93                | 71.70                | 64.94                | 64.73                | 65.70                |
> | $\alpha=0.25$     | 71.27                | 71.79                | 71.73                | 45.78                | 52.14                | 55.79                |
>
>
> References:
> - Bingcong Li and Georgios Giannakis. Enhancing sharpness-aware optimization through variance suppression. NeurIPS, 2023.
> - Xie, Wanyun, Thomas Pethick, and Volkan Cevher. SAMPa: Sharpness-aware Minimization Parallelized. NeurIPS, 2024.
> - Christina Baek, Zico Kolter, and Aditi Raghunathan. Why is sam robust to label noise? ICLR, 2024.
>
> *Conclusion*
>
> We thank you for your valuable feedback. We believe we have addressed all your concerns and have added experiments related to SANER's hyperparameters in **Appendix C**. As a result, we would really appreciate it if you consider increasing your score.

---

> ### Author Response · Authors · 2024-11-23
> **Kind Follow-Up Reminder**
>
> Dear Reviewer,
>
> Thank you once again for reviewing our manuscript. We have done our best to address your comments and have revised the paper based on suggestions from all reviewers.
>
> We understand today is the weekend and do not wish to disturb you. However, with the approaching deadline, we hope you could review the revisions and provide your feedback at your earliest convenience.
>
> Thank you for your valuable insights and support. Please let us know if you have any additional questions or need further clarification.

---

> > ### Comment · Reviewer_t7xA · 2024-11-24
> >
> > Thank you for your response, especially for adding the additional experiments on the α parameter. Most of my questions have been answered, and the newly added ablation experiments are valuable.
> >
> > However, I still recommend adding a section for formula derivation and proof to conduct a more in-depth theoretical exploration of the different values of α and k, similar to what Baek et al. (2024) did. Limiting the analysis to experimental results and using them to infer the validity of your hypothesis that "Group B explicitly controls noisy fitting" is still insufficient to fully convince me of the general applicability of the SANER method and the setting of the hyperparameters α and k.
> >
> > Therefore, I will maintain my score.

---

> > > ### Author Response · Authors · 2024-11-24
> > >
> > > Thank you once again for your insightful feedback. We truly appreciate your suggestion to incorporate theoretical analysis to further strengthen our proposed method.
> > >
> > > While we recognize the limitations of relying solely on empirical results, our extensive experiments and ablation studies provide valuable insights into SAM's behavior. These insights are both novel and innovative as noted in **your review** and those of **Reviewer CNqk**, **Reviewer 1dZ1**, and **Reviewer 3y6C**. Our proposed method achieves highly competitive results compared to SGD and SAM across various scenarios and demonstrates robustness when integrated with SAM-like variants. This evidence strongly supports the validity of our findings and the general applicability of the SANER method.
> > >
> > > We kindly hope that our work can be reviewed from the perspective of an empirical study, as our contributions in this regard are significant and provide a strong foundation for future theoretical exploration. Thank you again for your thoughtful engagement with our work.

---

### Official Review · Reviewer_3y6C · 2024-11-02

**Soundness:** 3
**Presentation:** 3
**Contribution:** 2
**Rating:** 5
**Confidence:** 5

**Summary:**

The paper conducts an empirical study which reveals that during training, some gradient dimensions contribute to the superior robustness of SAM over SGD against noisy labels. These dimensions have lower magnitudes and the same signs as the corresponding components in SGD. Reducing the weight of such dimensions during training improves robustness of SGD and SAM variants against label noise.

**Strengths:**

The paper reveals an interesting phenomena, showing that there are dimensions with lower magnitudes and the same signs as the corresponding components in SGD, that are responsible for superior robustness of SAM. This is interesting and novel to my knowledge.

The paper is clear and easy to follow. While I found the observation interesting, I believe the paper requires to back up the results with a more in-depth study (either some theoretical analysis or more in-depth ablation study and a wider range of experiments) to be considered significant.

**Weaknesses:**

While the proposed idea of reweighting dimensions with lower magnitudes and same signs as SGD is simple (simplicity is good), it requires an appropriate linear scheduler that gradually decreases the reweighting factor from 1 to a predetermined value over k epochs. Both the reweighting factor and k needs to be tuned and I didn't find an ablation study or any insight on how to set these parameters for different datasets/architectures. Table 5, 6 in the appendix only compares using or not using this scheduler (k=0 and k=50).

Considering that this is an empirical study, I'd expect much more in-depth empirical study of the observed phenomena. For example, what is the effect of datasets and architectures on the observed phenomena? What's the effect of the label noise level on the observed phenomena? How should one set the parameters of the proposed method? The paper includes experiments on Cifar10-100 (and their subsampled versions) with label noise level of 25% and 50%, and Webvision. On Cifar-10, the improvement is not significant, while on Cifar-100 the proposed method works much better. Nevertheless, there is no explanation or deeper analysis provided in the paper. Existing methods for robust training against noisy labels usually perform much better on Cifar10. Considering that the pattern is different for the proposed method, digging deeper into why things work differently would be an interesting addition to the paper. The authors can also analyze the effect of the proposed reweighting methods combined with existing robust training methods, as is done in the original SAM paper.

Finally, I believe adding a theoretical study of the observed phenomena using a simple data model and network architecture, or even a simple toy example explaining the effect of reweighting would significantly strengthen the paper.

---------------------------------

After the rebuttal:
I thank the authors for trying to address my concerns and in particular for providing additional experimental results, in particular on tinyimagenet with different architectures, adding experiments with robust methods, and sensitivity analysis. I also read the other reviews and authors responses. While, I believe that there are merits to the proposed method, I still find the paper lacking explanation on "why the the observed phenomena happens". To address this, the paper would significantly benefit from some (even simple) theoretical analysis (or even example). Without this, authors statement that their proposal "is not tailored to any specific dataset or architecture" looks somewhat over-claimed. Even a simple theoretical example using e.g. sparse coding data models and probably linear models can help clarifying why the observed phenomena can occur in more general setting. I have increased my score to acknowledge authors' effort in addressing the reviewers' concerns (mine and others). Nevertheless, I still don't think the paper is ready for publication at a top-tier ML conference.

**Questions:**

Please see the weaknesses above.

---

> ### Author Response · Authors · 2024-11-21
> **Thank you for your time and constructive feedback. We have addressed your concern. (Part 1/3)**
>
> >*Both the reweighting factor and k needs to be tuned and I didn't find an ablation study.*
>
> We used $\alpha=0.5$ and $k=50$ for all experiments across various architectures and datasets, demonstrating that achieving superior performance compared to SAM does not require extensive tuning. We ran experiments for $\alpha \in \{0.25, 0.5, 0.75\}$ and $k \in \{25, 50, 75\}$, trained on ResNet34 and WideResNet40-2 with CIFAR-100 under 25% and 50% noisy labels, as shown in Table.
>
> Key Observations:
> 1. **Sensitivity to $k$:**
> - Performance is relatively stable across different values of $k$ when $\alpha$ is moderate (0.5 or 0.75). This indicates that our method is not highly sensitive to $k$ under well-performing settings.
> - **In underfitting scenarios** (e.g., when $\alpha$ is small or noise is high), the choice of $k$ becomes critical. Higher values of $k$ (e.g., $k=75$) improve gradient reweighting, enhancing the model’s ability to fit clean data effectively and preventing severe underfitting.
> 2. **Sensitivity to $\alpha$:**
> - When $\alpha$ is too small (e.g., 0.25), the model struggles to fit clean data effectively, especially under higher noise levels (50%).
> - Moderate values of $\alpha$ (0.5 or 0.75) provide a good balance, achieving strong clean fitting and noise resistance without requiring extensive tuning.
>
>
>
> |  | Noise = 25% ($k=25$) | Noise = 25% ($k=50$) | Noise = 25% ($k=75$) | Noise = 50% ($k=25$) | Noise = 50% ($k=50$) | Noise = 50% ($k=75$) |
> |-------------------|----------------------|----------------------|----------------------|----------------------|----------------------|----------------------|
> | **RN34**      |                      |                      |                      |                      |                      |                      |
> | $\alpha=0.75$     | 73.09                | 72.76                | 72.04                | 66.37                | 66.28                | 64.73                |
> | $\alpha=0.5$      | 73.96                | 74.03                | 74.01                | 66.93                | 67.17                | 66.73                |
> | $\alpha=0.25$     | 74.33                | 74.49                | 74.25                | 61.51                | 66.60                | 67.48                |
> | **WRN40-2** |                      |                      |                      |                      |                      |                      |
> | $\alpha=0.75$     | 70.34                | 70.90                | 70.48                | 63.80                | 64.08                | 63.35                |
> | $\alpha=0.5$      | 71.55                | 71.93                | 71.70                | 64.94                | 64.73                | 65.70                |
> | $\alpha=0.25$     | 71.27                | 71.79                | 71.73                | 45.78                | 52.14                | 55.79                |
>
>
>
> >*I didn't find any insight on how to set these parameters for different datasets/architectures. Table 5, 6 in the appendix only compares using or not using this scheduler (k=0 and k=50). How should one set the parameters of the proposed method?*
>
> We believe that tuning hyperparameters based on training dynamics is a more robust and common practice compared to relying on datasets or architectures. To make the tuning process deterministic, the effect of hyperparameters on training dynamics must remain consistent across different setups.
>
> In our work, we introduced two hyperparameters, $\alpha$ and $k$, and analyzed their impact on training dynamics, as illustrated in Figure 5. To further verify their consistency, we conducted experiments varying $\alpha \in \{0.25, 0.5, 0.75\}$ and $k \in \{25, 50, 75\}$, evaluating them across different architectures. Noisy train accuracy across different $\alpha$ is shown in [this Figure](https://imgur.com/KQucMmN) and clean accuracy across different $k$ is shown in [this Figure](https://imgur.com/isQd3kN). Our experiments indicate that the characteristics of $\alpha$ and $k$ remain consistent across various configurations.
>
> Based on these observations, we provide guidelines for tuning these parameters according to training dynamics:
> - *$\alpha$ (Reweighting Factor):* The $\alpha$ parameter is directly proportional to the noisy fitting rate, as demonstrated in Figure 5a. Lower $\alpha$ values increase resistance to noisy label fitting, making it advantageous to reduce $\alpha$ when the model shows signs of *overfitting*.
> - *$k$ (Linear Scheduler):* The $k$ parameter governs the clean fitting rate during the early stages of training. If the model struggles to fit clean samples effectively—resulting in *underfitting* where training accuracy is low—increasing $k$ can help prevent this issue by improving clean sample fitting.

---

> ### Author Response · Authors · 2024-11-21
> **Thank you for your time and constructive feedback. We have addressed your concern. (Part 2/3)**
>
> >*What is the effect of datasets and architectures on the observed phenomena? What's the effect of the label noise level on the observed phenomena?*
>
> The main phenomena observed in our paper are as follows:
> 1. The experiment reveals that SAM not only up-weights gradients but also involves a significant portion of down-weighted gradients.
> - We demonstrate in [this Figure](https://imgur.com/YJnLzqO) that Groups A and B account for a larger proportion than Group C during the early stages of training across different architectures.
> 2. Group B primarily contributes to noisy fitting (Section 3).
> - We demonstrate that Group B mainly contribute to prevent noisy fitting in ResNet18, ResNet34, ResNet18 with doubled width in each layer. Group B contribute on par with Group A to prevent noisy fitting in WideResNet40-2 and DenseNet121. Our experiments, conducted on CIFAR-10 with 25% and 50% label noise, are shown in [this Figure](https://imgur.com/pVsT0O5).
> - We did not present this phenomenon on CIFAR-100 because it is challenging to illustrate the difference in noisy fitting between SAM and SGD, as indicated by the noisy training accuracy in Figure 17 of our revised manuscript. However, this figure also highlights that the reweighting of Group B (our method) remains effective in slowing down the noisy fitting rate.
> 3. Group B has a greater impact on slowing down noisy fitting compared to clean fitting (Section 4).
> - We also show that Group B consistently has a greater impact on slowing down noisy fitting compared to clean fitting across different architectures and datasets, including ResNet, WideResNet, and DenseNet trained on CIFAR-10 and CIFAR-100 with 25% and 50% label noise, as shown in [this Figure](https://imgur.com/k1C32PC).
>
> >*On Cifar-10, the improvement is not significant. While on Cifar-100 the proposed method works much better. Nevertheless, there is no explanation or deeper analysis provided in the paper.*
>
> The improvement of our method on CIFAR-10 is **significant**. Without noisy labels, SAM achieves a best performance of approximately 96% with ResNet18 as shown in Table. However, its performance drops to 93% when noisy labels are present, resulting in a 3% performance gap. In contrast, our method, SANER, achieves 94%, reducing the performance gap to 2%. This represents around 33% improvement in closing the gap compared to SAM trained on clean CIFAR-10.
>
> The performance on CIFAR-100 exhibits **a similar pattern** to the improvements observed on CIFAR-10. The performance of SAM without noisy labels is 79% as we ran and shown in Table. However, with 25% noisy labels, SAM's performance drops to 70%, resulting in a 9% gap. In comparison, SANER achieves 73%, reducing the gap to 6%. This also represents around 33% improvement.
>
> | Dataset    | Noise | SAM              |
> |------------|-------|------------------|
> | CIFAR-10   | 0%    | 96.04 ± 0.04     |
> | CIFAR-100  | 0%    | 79.19 ± 0.22     |
>
> Our proposed method is based on the behavior of SAM and is not tailored to any specific dataset or architecture. We demonstrated the consistent improvement of SANER over SAM across CIFAR-10, CIFAR-100, and Mini-Webvision, as shown in our paper, as well as on Tiny-ImageNet, which we provide for additional reference.
>
> | Dataset         | Architecture      | Param  | SGD   | SAM   | SANER                                     |
> |-----------------|-------------------|--------|-------|-------|-------------------------------------------|
> | **Tiny-ImageNet**| ResNet18          | 11.2M  | 56.50 | 57.60 | **61.60** (+4.00)     |
> |                 | ResNet34          | 21.3M  | 56.82 | 59.30 | **63.22** (+3.92)     |
> |                 | WideResNet28-10   | 36.5M  | 57.94 | 59.84 | **64.08** (+4.24)     |

---

> ### Author Response · Authors · 2024-11-21
> **Thank you for your time and constructive feedback. We have addressed your concern. (Part 3/3)**
>
> >*The authors can also analyze the effect of the proposed reweighting methods combined with existing robust training methods, as is done in the original SAM paper.*
>
> We conducted experiments to compare the test accuracy of SAM and SANER trained on ResNet-18 with CIFAR-10 and CIFAR-100, integrated with hard bootstrapping (Reed et al., 2014), as shown in Table. The results showed that SANER consistently outperformed Bootstrap + SAM.
>
> | Dataset    | Noise | SAM    | SANER  | Bootstrap + SAM | Bootstrap + SANER |
> |------------|-------|--------|--------|-----------------|-------------------|
> | **CIFAR-10** | 25%   | 93.25  | 94.14  | 93.26           | **94.25**         |
> |             | 50%   | 88.77  | **90.65** | 89.33           | 90.64             |
> | **CIFAR-100**| 25%   | 69.60  | 72.78  | 71.01           | **73.45**         |
> |             | 50%   | 61.06  | **65.73** | 64.21           | 65.57             |
>
>
> References:
> - Reed, Scott, Honglak Lee, Dragomir Anguelov, Christian Szegedy, Dumitru Erhan, and Andrew Rabinovich. Training deep neural networks on noisy labels with bootstrapping. ICLR, 2015.
>
> *Conclusion*
>
> We thank you for your valuable feedback. We believe we have addressed all your concerns and have added experiments in Appendix B, C, and D.2. As a result, we would really appreciate it if you consider increasing your score.

---

> ### Author Response · Authors · 2024-11-23
> **Kind Follow-Up Reminder**
>
> Dear Reviewer,
>
> Thank you once again for reviewing our manuscript. We have done our best to address your comments and have revised the paper based on suggestions from all reviewers.
>
> We understand today is the weekend and do not wish to disturb you. However, with the approaching deadline, we hope you could review the revisions and provide your feedback at your earliest convenience.
>
> Thank you for your valuable insights and support. Please let us know if you have any additional questions or need further clarification.

---

> ### Comment · Reviewer_3y6C · 2024-11-25
> **Thanks for your response**
>
> I thank the authors for taking the time to addressing my and other reviewers concern. I've augmented my review and increased my score, but I still believe that the paper needs a major revision to be ready for publication.

---

> > ### Author Response · Authors · 2024-11-26
> >
> > We sincerely thank you for your thoughtful feedback and for acknowledging our efforts to address the concerns raised during the review process. We recognize that fully addressing the question of "Why does the observed phenomenon happen?" (or specifically, why *Group B* affects the *noisy fitting rate*) would provide deeper insight into our findings. However, this question is *non-trivial* and would likely require substantial *theoretical exploration* beyond the current scope of our work.
> >
> > Given the complexity of directly answering this question, we instead focused on addressing two **closely related** and **foundational questions**:
> > 1. **How** does Group B affect clean learning and noisy learning?
> >    In **Section 4**, we demonstrate that **Group B** much more impacts *noise-dominated gradients* comparing with clean-dominated gradients, indicating a bias in Group B towards fitting noisy labels.
> >
> > 2. **What happens if** we adjust Group B’s effect on the observed phenomenon?
> >    This question is addressed through our proposed method, which consistently **outperforms SAM** and **significantly resists the noisy fitting rate**, demonstrating the **practical validity** of the correlation between **Group B** and the **noisy fitting rate**.
> >
> > To conclude, these **empirical experiments** strongly support the **validity of the observed phenomenon** and highlight its potential for further exploration. While we agree that deeper theoretical analysis is essential for fully understanding the mechanisms involved, we believe our findings **provide a solid foundation and practical evidence for future investigation.**
> >
> > Once again, we sincerely appreciate your feedback, which has greatly contributed to the **clarity** and **scope** of our paper. Thank you for your time and for considering our work.

---

### Official Review · Reviewer_1dZ1 · 2024-11-05

**Soundness:** 3
**Presentation:** 3
**Contribution:** 2
**Rating:** 5
**Confidence:** 3

**Summary:**

The author studied the relationship between the gradients of the SAM and SGD algorithms, and through experimental validation, found a method more suitable for enhancing the model's resistance to overfitting on noisy label data. The algorithm modifies the weights of different parameter gradients during model updates to improve the model's robustness. In terms of experiments, the author conducted validation on several classic public datasets.

**Strengths:**

1. In the context of increasingly large training datasets, it is practically significant to effectively prevent models from overfitting on noisy label data. Relying solely on algorithms and rules to clean the data is often too costly, so optimizing algorithms to reduce the impact of noisy label data is more valuable.
2. The author's approach of comparing the relative sizes of the gradients of each parameter in the SAM and SGD algorithms during model updates, and using this as a criterion to assess whether the parameters are overfitting to the model due to noisy label data, is straightforward and exhibits a certain level of innovation. The experimental design is also very reasonable.
3. The author's experimental design is quite rigorous, and they conducted a substantial number of experiments and tests on several public datasets.

**Weaknesses:**

1.I noticed that Figure 2 in Section 3 is based on training a ResNet-18 model on CIFAR-10. Shouldn't testing be conducted on more models and datasets to confirm that the three types of parameters, Group A, B, and C, indeed exhibit the proportions mentioned in the paper?Perhaps conducting experiments on a larger ResNet model (ResNet-50) and a larger dataset like CIFAR-100 or ImageNet (if conditions permit) would provide more convincing results.

2.Based on the author's analysis, in fact, at epoch 100, there was a noticeable increase in the parameter ratio of Group C, which maintained around 10% during the early training stages. Would it be more meaningful to directly use the reverse gradients for noisy label data? Also, I noticed that the ratios of Group B and Group C are quite similar in the early stages of training, and the values for Group A are mostly around 1.0 while those for Group B are around 0.99. If this is the case, I don't believe that SGD and SAM have different perspectives on the parameters in these two groups.As before, I believe it is necessary to validate this distribution pattern on larger models and datasets. Additionally, I think it would be meaningful to further investigate whether there are significant changes in the parameters within the three groups as training progresses, as well as the specific distribution of values.

**Questions:**

Please refer to the comments on weaknesses.

---

> ### Author Response · Authors · 2024-11-21
> **Thanks for your review. We have addressed your concern.**
>
> >*I noticed that Figure 2 in Section 3 is based on training a ResNet-18 model on CIFAR-10. Shouldn't testing be conducted on more models and datasets to confirm that the three types of parameters, Group A, B, and C, indeed exhibit the proportions mentioned in the paper? Perhaps conducting experiments on a larger ResNet model (ResNet-50) and a larger dataset like CIFAR-100 or ImageNet (if conditions permit) would provide more convincing results.*
>
> Yes, the trends of Groups A, B, and C are consistent with our observed phenomenon when training ResNet50 on CIFAR-100, as shown in [this Figure](https://imgur.com/1Vtuzd6). During the early training, Groups A and B represent a larger proportion compared to Group C. The parameter distribution of Group C increases in the later stages of training, after the model has fit the noisy labels as shown in [this Figure](https://imgur.com/qGD7Gej).
>
> >*Based on the author's analysis, in fact, at epoch 100, there was a noticeable increase in the parameter ratio of Group C, which maintained around 10% during the early training stages. Would it be more meaningful to directly use the reverse gradients for noisy label data?*
>
> We mentioned it in lines 195–202 of our paper. "Analyzing Group C is particularly challenging due to the inconsistency between the objectives of SAM and SGD. The divergence in gradient component directions complicates the learning process, as reversed gradients may hinder effective learning from the data. Moreover, our study focuses on the “memorization” phase (Arpit et al., 2017), where the transition from fitting clean examples to overfitting noisy examples occurs. This typically happens during the middle stage of training, when most clean examples have already been learned. During this period, Groups A and B are still dominant compared to Group C. Therefore, in this section, we focus on comparing the effects of Group A and Group B of SAM on noisy fitting, leaving the analysis of Group C for future work."
>
> We would like to emphasize that the "noticeable increase in the parameter ratio of Group C, which maintained around 10% during the early training stages" is not a special phenomenon that occurs due to fitting to noisy labels. Even in clean data, Group C exhibited a similar pattern, as shown in [this Figure](https://imgur.com/is78BxD). This suggests that the increase in Group C might not be directly related to noisy label data.
>
>
> The motivation for focusing on Group B is grounded in two key points:
> - Group A, which comprises the largest subset of parameters, has been extensively discussed by Baek et al. (2024) in how upweighting gradients can help mitigate noisy fitting *implicitly*.
> - Group B represents the next significant subset of parameters and is characterized by its nature of slowing convergence. This observation leads to the hypothesis that this slower convergence may be linked to a reduced rate of learning noisy labels. Consequently, our analysis prioritizes Group B to investigate this relationship further.
> - To strengthen our finding in Figure 3 by including Group C, we ran the SGD-GrC experiment, which replaces the gradient of Group C in SAM with those from SGD, as shown in [this Figure](https://imgur.com/oKATPGC). Group B still mainly contributes to noisy fitting.
>
>
> >*Also, I noticed that the ratios of Group B and Group C are quite similar in the early stages of training, the values for Group A are mostly around 1.0 while those for Group B are around 0.99. If this is the case, I don't believe that SGD and SAM have different perspectives on the parameters in these two groups.*
>
> We illustrate the distribution of the ratios of the SAM gradient to the SGD gradient at the 60th, 120th, and 180th epochs in [this Figure](https://imgur.com/Aocj5j3) for your reference. The range of values for the ratios is wide and not concentrated mostly around 0.99 and 1.0, as you believe, indicating that Group A, Group B are very different from each other during training.
>
> References:
> - Pierre Foret, Ariel Kleiner, Hossein Mobahi, and Behnam Neyshabur. Sharpness-aware minimization for efficiently improving generalization. ICLR, 2021.
> - Christina Baek, Zico Kolter, and Aditi Raghunathan. Why is sam robust to label noise? ICLR, 2024.
>
> *Conclusion*
>
> We thank you for your valuable feedback. We believe we have addressed all your concerns and have added experiments related to the proportion of each group in **Appendix B.1**. As a result, we would really appreciate it if you consider increasing your score.

---

> ### Author Response · Authors · 2024-11-23
> **Kind Follow-Up Reminder**
>
> Dear Reviewer,
>
> Thank you once again for reviewing our manuscript. We have done our best to address your comments and have revised the paper based on suggestions from all reviewers.
>
> We understand today is the weekend and do not wish to disturb you. However, with the approaching deadline, we hope you could review the revisions and provide your feedback at your earliest convenience.
>
> Thank you for your valuable insights and support. Please let us know if you have any additional questions or need further clarification.

---

### Official Review · Reviewer_CNqk · 2024-11-09

**Soundness:** 2
**Presentation:** 4
**Contribution:** 2
**Rating:** 5
**Confidence:** 4

**Summary:**

This paper builds on the observation that SAM improves robustness to label noise over SGD. The paper analyzes which parameters' gradients are affected in different ways, between the SAM and SGD update. This provides some insight into the importance of further decreasing the norms of those parameters that have a lower norm under the SAM perturbation. Empirical results seem to show gains when performing this modification of SAM.

**Strengths:**

This paper builds up and contributes to the literature on understanding the benefits of SAM. The paper is generally well written and easy to read. Understanding SAM is broadly conceptually interesting, and the application to label noise is practically well motivated.

It builds up ideas in a simple fashion, and lays out clear hypotheses being tested and implications of these hypotheses. The study of which parameters have their gradients upweighted, down-weighted or reversed is novel to the best of my knowledge. The broader implications of this analysis are a little unclear (detailed below), but the analysis is clean and clearly presented.

The paper provides a new and simple modification to SAM based on the analysis and performs a large-scale evaluation over many datasets and architectures. While I have some reservations about the soundness of some aspects, overall this was a clear effort for a thorough investigation.

**Weaknesses:**

One glaring gap in the analysis of the paper is the effect of early-stopping. Early-stopping seems to improve clean test accuracy in both SAM and SGD. The paper does not make it clear whether the goal is to compare SANER, SAM, and SGD at the final checkpoint or best early stopped checkpoint. Even with early stopping, SAM outperforms SGD. However, from Fig 1(c) it seems like SAM vs SANER shows no gains if we do early stopping. In all the experiments reported later, it is unclear which checkpoints are compared.

From an analysis perspective, the authors acknowledge the weakness of focusing on noisy train accuracy (rather than final test accuracy): intuitively fitting noise at training is bad, but for different algos, this could manifest differnetly in terms of affecting test accuracy. Furthermore, clean training might also be affected which can affect test accuracy. I agree with the authors on this weakness.

From a practical perspective, if the gains do not hold with early stopping, the benefit is less clear. Furthermore, SANER introduces a new hyperparameter \alpha and adds complexity in practice.

The role of the perturbation radius of SAM is not discussed - in practice, this is another important parameter. How does this radius interact with the new hyperparameter \alpha introduced? Is it strictly better to tune \alpha rather than the radius, or should we tune both in parallel, or do they interact in ways such that we need to tune over both in combination.

Overall, the analysis is potentially clean and interesting, but for reasons above, the soundness / comprehensive analysis is missing some key aspects. There is no clear generalizable insights from this work, and gains in practice (if they hold up under further experiments) come at the cost of additional hyperparameters.

**Questions:**

(1) Please explain how you consider the role of early stopping, and what regime does the analysis and SANER experiments consider.

(2) Does SANER improve over SAM even accounting for early stopping, i.e. is the best early-stopped checkpoint of SANER better than best early-stopped checkpoint with SAM?

(3) I am sorry if I missed this, but how do you set the SAM perturbation radius hyperparameter. How does it interact with \alpha? Do the trends of gradients across different groups hold for different radii as well?

---

> ### Author Response · Authors · 2024-11-21
> **Thanks for your review. We have addressed your concern.**
>
> >*Please explain how you consider the role of early stopping, and what regime does the analysis and SANER experiments consider.*
>
> In our experiments, we report the best test accuracy checkpoint across all epochs and this approach is better than early stopping, this is because it effectively captures the model’s performance without carefully tuning hyperparameter patience of early stopping and matches training default in practice when we do not know training dataset containing noisy labels or not.
>
> >*Does SANER improve over SAM even accounting for early stopping, i.e. is the best early-stopped checkpoint of SANER better than best early-stopped checkpoint with SAM?*
>
> Yes, SANER performs better than SAM when using the best early-stopped checkpoint setting. You can refer to the test accuracy in Figures 1c, and 17 of our revised manuscript. The test accuracy of SANER is consistently higher than that of SAM across most epochs, which can include the best early-stopped checkpoint.
>
> >*How do you set the SAM perturbation radius hyperparameter?*
>
> We set the SAM perturbation radius hyperparameter $\rho = 0.1$ as the recommendation in the original SAM paper by Foret et al. (2021) for label noise settings.
>
> >*How does perturbation radius interact with $\alpha$?*
>
> We provide the Table compares the test accuracy of SAM and SANER under different radii $\rho = \{ 0.05, 0.1, 0.15, 0.2 \}$ for your reference. Tuning $\rho$ can slightly improve the performance of SANER, but with the default settings of $\alpha=0.5$ and $k=50$, SANER consistently outperforms SAM without being highly sensitive to the perturbation radius.
>
> | Dataset     | Opt    | $\rho = 0.05$ | $\rho = 0.1$  | $\rho = 0.15$ | $\rho = 0.2$  |
> |-------------|---------|-----------|----------|----------|----------|
> | **CIFAR-10** | SAM     | 91.93     | 93.25    | 93.46    | 93.71    |
> |             | SANER   | **93.32** | **94.18**| **94.09**| **94.23**|
> | **CIFAR-100**| SAM     | 68.33     | 69.60    | 70.30    | 70.86    |
> |             | SANER   | **72.57** | **72.78**| **73.15**| **73.72**|
>
>
> >*Do the trends of gradients across different groups hold for different radii as well?*
>
> Yes, these trends are consistent across different radii. We ran ResNet18 on CIFAR-10 with $\rho = \{ 0.05, 0.1, 0.15, 0.2 \}$ and show distribution of these groups in [this Figure](https://imgur.com/DoJIuj8). Groups A and B account for a larger proportion compared to Group C. The parameter distribution of Group C increases during the later stages of training when the model has already fit the noisy labels.
>
> References:
> - Pierre Foret, Ariel Kleiner, Hossein Mobahi, and Behnam Neyshabur. Sharpness-aware minimization for efficiently improving generalization. ICLR, 2021.
>
> *Conclusion*
>
> We thank you for your valuable feedback. We believe we have addressed all your concerns and have added experiments related to the perturbation radius in **Appendix B.1 and C.1**. As a result, we would really appreciate it if you consider increasing your score.

---

> ### Author Response · Authors · 2024-11-23
> **Kind Follow-Up Reminder**
>
> Dear Reviewer,
>
> Thank you once again for reviewing our manuscript. We have done our best to address your comments and have revised the paper based on suggestions from all reviewers.
>
> We understand today is the weekend and do not wish to disturb you. However, with the approaching deadline, we hope you could review the revisions and provide your feedback at your earliest convenience.
>
> Thank you for your valuable insights and support. Please let us know if you have any additional questions or need further clarification.

---

> > ### Comment · Reviewer_CNqk · 2024-12-01
> > **Response to rebuttal**
> >
> > Thank you for your rebuttal. I apologize for my delay on getting back to this.
> >
> > (1) Thank you for changing things to focus on best test accuracy rather than the last as I think this is a very important point. However, I distinctly remember that fig 1c from your initial submission did not show SANER outperforming SAM when considering early stopping. I tried but was unable to access the submitted PDF. Could you please clarify whether the figure was updated / did it show different models or different settings? It is possible I am misremembering but I just wanted to clarify.
> >
> > (2) Thank you for running experiments with additional perturbation radii - that is helpful! In both datasets, the largest radius seems to perform the best - makes me wonder if we should widen the hyperparameter sweep to get a true picture of how SAM and SANER differ.
> >
> > I read your response, the submission and the other reviews. Unfortunately, I believe this paper is borderline and would still lean towards rejection for the following reasons.
> >
> > - The method, while simple, does introduce additional complexity that makes it harder for the method to become a standard practice
> > - The contribution feels a little incremental because it is not clear what the general insights are. The intuition and results feel a little clunky as evidenced by the questions asked by all the reviewers.
> >
> > That all said, I do appreciate the efforts by the authors and the overall findings are interesting. I would not be upset if this paper is accepted, but I wouldn't push for acceptance.

---

### Author Response · Authors · 2024-11-21
**Rebuttal Revision**

We thank the reviewers for their encouraging comments and useful insights. We have made the following changes in the newly uploaded version (and will continue to improve the paper):
- **Gradient behavior analysis**: In response to concerns about providing more evidence regarding the gradient behavior analyzed in our paper, we conducted new experiments across various settings. The results, presented in Appendix B, support the observed phenomena.
- **Ablation study for SANER hyperparameters**: In response to concerns about the ablation study for SANER hyperparameters, we performed additional experiments exploring different settings, including various values of $\rho$, $\alpha$, and $k$. These results, shown in Appendix C, demonstrate that SANER's hyperparameters exhibit consistent behavior, aligning with our analysis in Section 5.
- **SANER performance across diverse setups**: Addressing concerns about SANER's performance in more diverse setups, we evaluated it under various noise rates (including noise-free scenarios), integration with the Bootstrapping method, and on large datasets. These results, presented in Appendix D, demonstrate that our method consistently outperforms SAM.

---

### Meta-Review · Area_Chair_Xcym · 2024-12-22

**Metareview:**

The work concerns learning classification models under training label noise. It builds on the recently proposed optimization for better generalization, Sharpness-aware minimization (SAM), which has also been shown to have robustness to label noise. It modifies SAM, by first decomposing parameters based on their gradient’s sign and magnitude when using SGD and SAM, then empirically observing SAM’s gradient for some parameters contribute to noise robustness/fitting effect, and then reweighting the components that are considered to have the robustness effect. The new optimization procedure is called SANER. The paper tests SANER on CIFAR datasets as well as mini-WebVision and shows improved results over standard SAM.

The reviewers appreciated the interesting observation regarding the different parameter gradients’ contribution in robustness, the paper taking a further step in understanding SAM, the novelty of the view in analyzing the parameter gradients, clarity and accessibility of the writing and presentations, and the extent of architectures and datasets for empirical evaluations,

On the other hand the reviewers raised concerns regarding lack of enough ablation studies for an empirical paper including the effect of early stopping, effect of SAM and SANER hyperparameters individually and in conjunction, strength of the empirical evidence for the difference between the different groups when using SGD, SAM, and SANER, and the lack of a rigorous grouping of parameters and a formal insight into the groups and their effects.

The authors provided a rebuttal with clarifications and additional experiments. While some clarifications were helpful, the additional experiments did not convince the reviewers.

The AC believes the paper brings about an interesting observation which generally deserves publication with a rigorous study, however, the current paper being an empirical improvement of SAM for robustness to label noise, needs to put more effort into convincing the reader that SANER is in fact the objective to use in stead of SAM when training data contain label noise. As reviewers suggested this is still a question mark for several reasons for instance, with early stopping the effect seems to diminish or the number of hyperparameters increase and its sensitivity is not entirely clear to make it useful, among other concerns. In addition to this, a more thorough empirical study of these identified groups and their grouping criteria in different noise settings, dataset types, optimization types and parameters, should be done to better understand where and why the reweighting can help or not help.

Alternatively, or at the same time, the paper can become stronger in providing more formal or concrete insights into the noise-robustness of SAM which would also be a contribution. Currently, while it is quite interesting to know about the observations, the take-away insight/message seems not fully baked.

Therefore, overall, the AC agrees with the majority of the reviewers and believes the paper reads unfinished and needs a major revision to be accepted. That is despite being interesting and well-written. It should become a stronger submission if the points raised by the reviewers are taken into account for the next revision.

**Additional Comments On Reviewer Discussion:**

The paper was reviewed by a panel of five reviewers with expertise in robustness to label noise, optimization, and SAM. While reviewers rating make it a borderline paper, most reviewers lean on the reject side despite considering and discussing the authors’ rebuttal. The AC, after considering the paper, the reviews, the rebuttal, and the discussions, sides with the majority of the reviewers.

---

### Decision · Program_Chairs · 2025-01-22

Reject